# SHORTCUT LEARNING THROUGH THE LENS OF EARLY TRAINING DYNAMICS

## ABSTRACT

Deep Neural Networks (DNNs) are prone to learn *shortcut* patterns that damage the generalization of the DNN during deployment. Shortcut Learning is concerning, particularly when the DNNs are applied to safety-critical domains. This paper aims to better understand shortcut learning through the lens of the learning dynamics of the internal neurons during the training process. More specifically, we make the following observations: (1) While previous works treat shortcuts as synonymous with spurious correlations, we emphasize that not all spurious correlations are shortcuts. We show that shortcuts are only those spurious features that are "easier" than the core features. (2) We build upon this premise and use *instance difficulty* methods (like Prediction Depth (Baldock et al., 2021)) to quantify "easy" and to identify this behavior during the training phase. (3) We empirically show that shortcut learning can be detected by observing the learning dynamics of the DNN's *early layers*, irrespective of the network architecture used. In other words, easy features learned by the initial layers of a DNN early during the training are potential shortcuts. We verify our claims on simulated and real medical imaging data and justify the empirical success of our hypothesis by showing the theoretical connections between Prediction Depth and information-theoretic concepts like $\mathcal{V}$-usable information (Ethayarajh et al., 2021). Lastly, our experiments show the insufficiency of monitoring only accuracy plots during training (as is common in machine learning pipelines), and we highlight the need for monitoring early training dynamics using example difficulty metrics.

## 1 INTRODUCTION

*Shortcuts* are spurious features that perform well on standard benchmarks but fail to generalize to real-world settings (Geirhos et al., 2020). Deep neural networks (DNNs) tend to rely on shortcuts even in the presence of *core* features that generalize well, which poses serious problems when deploying them in safety-critical applications such as finance, healthcare, and autonomous driving (Geirhos et al., 2020; Oakden-Rayner et al., 2020; DeGrave et al., 2021). Previous works view shortcut learning as a distribution shift problem (Kirichenko et al., 2022; Wiles et al., 2021; Bellamy et al., 2022; Adnan et al., 2022). However, we show that not all spurious correlations are shortcuts. Models suffer from shortcut learning only when the spurious features are much easier to learn than signals that generalize well. We show how monitoring example difficulty metrics like Prediction Depth (PD) (Baldock et al., 2021) can reveal valuable insights into shortcut learning quite early during the training process. Early detection of shortcut learning is useful as it can help develop intervention schemes to fix the shortcut early. To the best of our knowledge, we are the first to detect shortcut learning by monitoring the training dynamics of the model.

Geirhos et al. (2020) define shortcuts as spurious correlations that exist in standard benchmarks but fail to hold in more challenging test conditions, like real-world settings. The emphasis on shortcuts being synonymous with spurious correlations has led to widespread adoption of viewing shortcut learning as a distribution shift problem (Bellamy et al., 2022; Wiles et al., 2021; Adnan et al., 2022; Kirichenko et al., 2022). While the distribution shift explains part of the story, we emphasize that what is equally important for shortcut learning is the difficulty of the spurious features themselves (see Fig-1). Previous works like Shah et al. (2020); Scimeca et al. (2021) hint at this. But we take this line of thought further by viewing shortcut learning as a phenomenon that impacts the dataset difficulty, which can be captured by monitoring early training dynamics.

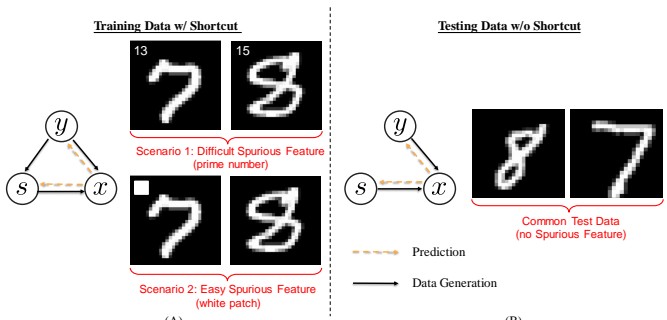

Figure 1: An illustration of how the causal view of shortcut learning is insufficient. In the causal view, training and testing are different graphical models between input ($x$), output ($y$), and the spurious feature ($s$). If $x$ can predict $s$, and $y$ is not causally related to s on the test data, then $s$ is viewed as a shortcut. (A) The figure shows two scenarios for even-odd classification. Scenario-1 shows a dataset where all even numbers have a spurious composite number (located at the top-left), and odd numbers have a prime number. Scenario-2 shows a dataset where all odd numbers have a spurious white patch. The spurious white patch is an easy feature, so the model uses it as a shortcut. Whereas classifying prime numbers, as shown in scenario 1, is challenging. So the model ignores such spurious features. This shows not all spurious correlations are shortcuts.

The premises that support our hypothesis are as follows: **(P1)** Shortcuts are only those spurious features that are *"easier"* to learn than the core features (see Fig-1). **(P2)** Initial layers of a DNN tend to learn easy features, whereas the later layers tend to learn the harder ones (Zeiler & Fergus, 2014; Baldock et al., 2021). **(P3)** Easy features are learned much earlier than the harder ones during training (Mangalam & Prabhu, 2019; Rahaman et al., 2019). Premises **(P1-3)** lead us to conjecture that: *"Easy features learned by the initial layers of a DNN early during the training are potential shortcuts."*

We empirically show that our hypothesis works well on simulated and real medical imaging data (section-4.2) regardless of the DNN architecture used. We justify this empirical success by theoretically connecting prediction depth with information-theoretic concepts like $\mathcal{V}$-usable information (Ethayarajh et al., 2021) (section-3 and appendix-A.1). Lastly, our experiments highlight that monitoring only accuracy during training, as is common in machine learning pipelines, is insufficient. Rather we need to monitor the learning dynamics of the model using suitable metrics to detect shortcut learning (section-4.3). This could potentially save a lot of time and computational costs and help develop reliable models that do not rely on spurious features.

## 2 RELATED WORK

**Not all spurious correlations are shortcuts:** Geirhos et al. (2020) define shortcuts as spurious correlations that exist in standard benchmarks but fail to hold in more challenging test conditions. Wiles et al. (2021) view shortcut learning as a distribution shift problem where two or more attributes are correlated during training but are independent in the test data. Bellamy et al. (2022) use causal diagrams to explain shortcuts as spurious correlations that hold during training but not during deployment. All these papers characterize shortcuts purely as a consequence of distribution shift; methods exist to build models robust to such shifts (Arjovsky et al., 2019; Krueger et al., 2021; Puli et al., 2022). In contrast, we stress that not all spurious correlations are shortcuts. Rather only those spurious features that are easier than the core features are potential shortcuts (see Fig-1). Previous works like Shah et al. (2020); Scimeca et al. (2021) hint at this by saying that DNNs are biased towards simple solutions, and Dagaev et al. (2021) use the "too-good-to-be-true" prior to emphasize that simple solutions are unlikely to be valid across contexts. Veitch et al. (2021) distinguish various model features using tools from causality and stress test the models for counterfactual invariance. Other works in natural language inference, visual question answering, and action recognition, also assume that simple solutions could be potential shortcuts (Sanh et al., 2020; Li & Vasconcelos, 2019; Clark et al., 2019; Cadene et al., 2019; He et al., 2019). We take this line of thought further by viewing shortcuts as simple solutions or, more explicitly, easy features, which affect the early training dynamics of the model. We suggest using suitable example difficulty metrics to measure this effect.

**Estimating Example Difficulty:** There are different metrics in the literature for measuring instance-specific difficulty (Agarwal et al., 2022; Hooker et al., 2019; Lalor et al., 2018). Jiang et al. (2020) train many models on data subsets of varying sizes to estimate a consistency score that captures the probability of predicting the true label for a particular example. Toneva et al. (2018) define example difficulty as the minimum number of iterations needed for a particular example to be predicted correctly in all subsequent iterations. Agarwal et al. (2022) propose a VoG (variance-of-gradients) score which captures example difficulty by averaging the pre-softmax activation gradients across training checkpoints and image pixels. Feldman & Zhang (2020) use a statistical viewpoint of measuring example difficulty and develop influence functions to estimate the actual leave-one-out influences for various examples. Ethayarajh et al. (2021) use an information-theoretic approach to propose a metric called pointwise $\mathcal{V}$-usable information (PVI) to compute example difficulty. Baldock et al. (2021) define prediction depth (PD) as the minimum number of layers required by the DNN to classify a given input and use this to compute instance difficulty. In our experiments, we use the PD metric to provide a proof of concept for our hypothesis.

**Monitoring Training Dynamics:** Other works that monitor training dynamics have a significantly different goal than ours. While we monitor training dynamics to detect shortcut learning, Rabanser et al. (2022) use neural network training dynamics for selective classification. They use the disagreement between the ground truth label and the intermediate model predictions to reject examples and obtain a favorable accuracy/coverage trade-off. Feng & Tu (2021) use a statistical physics-based approach to study the training dynamics of stochastic gradient descent (SGD). While they study the effect of mislabeled data on SGD training dynamics, we study the effect of shortcut learning on the early-time learning dynamics of DNNs. Hu et al. (2020) use the early training dynamics of neural networks to show that a simple linear model can often mimic the learning of a two-layer fully connected neural network. Adnan et al. (2022) have a similar goal as ours and use mutual information to monitor shortcut learning. However, computing mutual information is intractable for high-dimensional data, and hence their work is only limited to infinite-width neural networks that offer tractable bounds for this computation. Our work, on the contrary, is more general and holds for different neural network architectures and datasets.

## 3 BACKGROUND AND METHODOLOGY

In this section, we formalize the notion of spurious features, shortcuts, and task difficulty. Let $P_{tr}$ and $P_{te}$ be the training and test distributions defined over the random variables $\mathbf{X}$ (input), $\mathbf{y}$ (label), and $\mathbf{s}$ (*latent* spurious feature).

**Definition-1 (Spurious Feature s):** A latent feature $\mathbf{s}$ is called spurious if it is correlated with label $\mathbf{y}$ in the training data but not in the test data. Specifically, the joint probability distributions $P_{tr}$ and $P_{te}$ can be factorized as follows.

$$P_{tr}(\mathbf{X}, \mathbf{y}, \mathbf{s}) = P_{tr}(\mathbf{X}|\mathbf{s}, \mathbf{y})P_{tr}(\mathbf{s}|\mathbf{y})P_{tr}(\mathbf{y}) \tag{1}$$

$$P_{te}(\mathbf{X}, \mathbf{y}, \mathbf{s}) = P_{tr}(\mathbf{X}|\mathbf{s}, \mathbf{y})P_{te}(\mathbf{s})P_{tr}(\mathbf{y}). \tag{2}$$

The variable $\mathbf{s}$ appears to be causally related to $\mathbf{y}$ but is not. This is shown in Fig-1. We also need a notion of task difficulty. The difficulty of a task depends on the model and data distribution $(\mathbf{X}, \mathbf{y})$.

**Definition-2 (Task Difficulty $\Psi$):** Let $\Psi^P_{\mathcal{M}}(\mathbf{X} \to \mathbf{y})$ indicates the difficulty of predicting $\mathbf{X} \to \mathbf{y}$ for a model $\mathcal{M}$, where $\mathbf{X}, \mathbf{y} \sim P$. Consider a joint distribution $(\mathbf{X}, \mathbf{y}, \mathbf{t}) \sim P$ for two tasks, $\mathbf{t}$, and $\mathbf{y}$. Then, $\Psi^P_{\mathcal{M}}(\mathbf{X} \to \mathbf{y}) > \Psi^P_{\mathcal{M}}(\mathbf{X} \to \mathbf{t})$ indicates that the task $\mathbf{X} \to \mathbf{y}$ is harder than $\mathbf{X} \to \mathbf{t}$ for a given model $\mathcal{M}$.

**Definition-3 (Shortcut):** The spurious feature $\mathbf{s}$ is a potential shortcut for model $\mathcal{M}$ iff $\Psi^{P_{tr}}_{\mathcal{M}}(\mathbf{X} \to \mathbf{y}) > \Psi^{P_{tr}}_{\mathcal{M}}(\mathbf{X} \to \mathbf{s})$. In other words, given the input $\mathbf{X}$, predicting spurious feature $\mathbf{s}$ is easier for $\mathcal{M}$ than predicting the true label $\mathbf{y}$.

Note that we use the term "potential shortcut" because, in the presence of multiple spurious features, Scimeca et al. (2021) empirically show that the model selectively favors the easiest spurious feature.

We now explain two metrics (Prediction Depth and $\mathcal{V}$-Usable Information) to measure $\Psi_{\mathcal{M}}^P$. Baldock et al. (2021) proposed the notion of Prediction Depth (PD) to estimate example difficulty. The PD of input is defined as the minimum number of layers required by the model to classify the input. Our work relies on monitoring early training dynamics using example difficulty metrics, and we use PD estimation for this purpose. The PD metric is also well-suited for our work as it is defined for easy inputs even when the DNN is not fully trained. We use a binary classification setting to explain the concepts used in this section.

**Notion of Prediction Depth:** The PD is defined by building $k$-NN classifiers on the embedding layers of the model. The PD is simply the earliest layer after which all subsequent $k$-NN predictions remain the same (0 or 1):

$$\text{PD} = \min \arg \max_{n} \left[ \prod_{i=n}^{N} f_{knn}(\phi^i) + \prod_{i=n}^{N} (1 - f_{knn}(\phi^i)) \right], \qquad (3)$$

$f_{knn}$ is a $k$-NN classifier that outputs 0 or 1 based on a given threshold (see Appendix-A.4 for details), $\phi^i$ is the feature embedding for the given input at layer-$i$, and $N$ is the index of the final layer of the model. The lower the PD of input, the easier it is to classify. We also use the notion of undefined PD to work with models that are not fully trained. We treat $k$-NN predictions close to 0.5 (for a binary classification setting) as invalid. If the $k$-NN predictions for the last three layers (for a given input to the model) are invalid, we treat the PD of the input as undefined (see Appendix-A.4). Figure-2 illustrates how to read the PD plots used in our experiment.

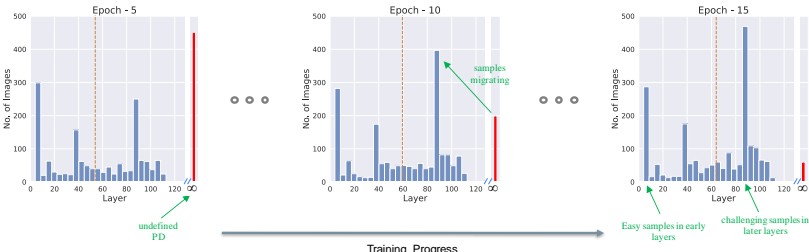

Figure 2: Examples of PD plots (for DenseNet-121) at different stages of the training process. The red bar indicates samples with undefined PD, and the dotted vertical line indicates the mean PD of the plot. Notice that the undefined samples (shown in red) slowly accumulate in layer 88 as training progresses. This is because the model needs more time to learn the challenging samples which accumulate at higher prediction depth, i.e., later layers.

**Notion of $\mathcal{V}$-Usable Information:** The Mutual Information between input and output, $I(X;Y)$, is invariant with respect to lossless encryption of the input, i.e., $I(\tau(X);Y) = I(X;Y)$. Such a definition assumes unbounded computation and is counter-intuitive to define task difficulty as heavy encryption of $X$ does not change the task difficulty. The notion of *"Usable Information"* introduced by Xu et al. (2020) assumes bounded computation based on the model family $\mathcal{V}$ under consideration. Usable information is measured under a framework called *predictive $\mathcal{V}$-information* (Xu et al., 2020). Ethayarajh et al. (2021) introduce *pointwise $\mathcal{V}$-information* (PVI) for measuring example difficulty.

$$\text{PVI}(x \rightarrow y) = -\log_2 g[\phi](y) + \log_2 g'[x](y), \qquad \text{s.t.} \qquad g, g' \in \mathcal{V} \qquad (4)$$

The function $g$ is trained on $(\phi, y)$ input-label pairs, where $\phi$ is a null input that provides no information about the label $y$. $g'$ is trained on $(x, y)$ pairs from the training data. Lower PVI instances are harder for $\mathcal{V}$ and vice-versa. Since the first term in Eq-4 corresponding to $g$ is independent of the input $x$, we only consider the second term having $g'$ in our experiments. In what follows, we relate the notions of PD and $\mathcal{V}$-usable information. We use $\mathcal{V}_{cnn}$ (of finite depth and width) in our proof as our experiments mainly use CNN architectures.

**Proposition 1:** (Informal) Consider two datasets: $D_s \sim P_{tr}(\mathbf{X}, \mathbf{y})$ with shortcuts and $D_i \sim P_{te}(\mathbf{X}, \mathbf{y})$ without shortcuts. For some mild assumptions on PD (see Appendix-A.1), if the mean

PD of $D_s$ is less than the mean PD of $D_i$, then the $\mathcal{V}_{cnn}$-usable-information for $D_s$ is larger than the $\mathcal{V}_{cnn}$-usable-information for $D_i$: $\mathcal{I}_{\mathcal{V}_{cnn}}^{D_s}(X \to Y) > \mathcal{I}_{\mathcal{V}_{cnn}}^{D_i}(X \to Y)$.

See proof in Appendix-A.1. The proposition intuitively implies that a sufficient gap between the mean PD of shortcuts and core features can result in shortcuts having more $\mathcal{V}_{cnn}$-usable information than core features. In such a scenario, the model will be more inclined to learn shortcuts over core features. This proposition justifies using the PD metric to detect shortcut learning, as demonstrated in the following experiments.

# 4 EXPERIMENTS

We set up four experiments to evaluate our hypothesis. *First*, we consider two kinds of datasets, one where the spurious feature is easier than the core feature and another where the spurious feature is harder. We train a classifier on each dataset and observe that the model can learn the easy spurious feature but not the harder one. This experiment demonstrates that not all spurious correlations are shortcuts, but only those spurious features that are easier than the core features are potential shortcuts. *Second*, we use toy and multiple medical datasets to demonstrate how monitoring the learning dynamics of the initial layers can reveal suspicious shortcut learning activity. We show that an early peak in the PD plot indicates potential shortcuts, and visualization techniques like grad-CAM can provide intuition about the feature used at that layer. *Third*, we show how shortcuts can often be detected relatively early during training. This is because initial layers which learn the shortcuts converge very early during the training. We observe this by monitoring PD plots across training epochs. In all of our experiments, the shortcut is revealed by the PD plot within two epochs of training. *Fourth*, we show that datasets with easy spurious features have more "usable information" (Ethayarajh et al., 2021) compared to their counterparts without such features. Due to higher usable information, the model requires fewer layers to classify the images with spurious features. We use this experiment to empirically justify Proposition-1 outlined in Appendix-A.1.

## 4.1 NOT ALL SPURIOUS CORRELATIONS ARE SHORTCUTS

We use the Dominoes binary classification dataset (formed by concatenating two datasets vertically; see Fig-4.1) similar to the setup of Kirichenko et al. (2022). The bottom (top) image acts as the core (spurious) feature. Images are of size $64 \times 32$. We construct three pairs of domino datasets such that each pair has both a hard and an easy spurious feature with respect to the common core feature (see Table-1). We use classes {0,1} for MNIST and SVHN, {coat,dress} for FMNIST, and {airplane, automobile} for CIFAR10. We also include two classes from Kuzushiji-MNIST (or KMNIST) and construct a modification of this dataset called KMNpatch, which has a spurious patch feature (5x5 white patch on the top-left corner) for one of the two classes of KMNIST. The spurious features are perfectly correlated with the target. The order of dataset difficulty based on the mean-PD is as follows: KMNpatch(1.1) < MNIST(2.2) < FMNIST(3.9) < KMNIST(5) < SVHN(5.9) < CIFAR10(6.8). We use a ResNet18 model and measure the test and core-only accuracies. The test accuracy is measured on a held-out dataset sampled from the same distribution. For the core-only accuracy, we blank out the spurious feature (top-half image) by replacing it with zeroes (same as Kirichenko et al. (2022)). The higher the core-only accuracy, the lesser the model's reliance on the spurious feature.

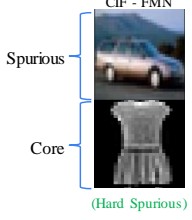
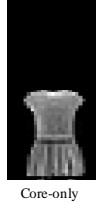

CIF - FMN   MN - FMN

Spurious

Core

(Hard Spurious)   (Easy Spurious)   Core-only Image

Figure 3: Dominoes Dataset

| Dataset (Spurious-Core) | Is spurious harder than core? | Test Accuracy | Core-only Accuracy |
|---|---|---|---|
| CIF(6.8) - FMN(3.9) | yes | 99.12±0.27% | 98.95±0.30% |
| MN(2.2) - FMN(3.9) | no | 99.95±0.05% | 50.75±2.96% |
| CIF(6.8) - KMN(5) | yes | 98.91±0.16% | 98.30±1.08% |
| MN(2.2) - KMN(5) | no | 99.97±0.05% | 50.48±2.64% |
| CIF(6.8) - MN(2.2) | yes | 99.74±0.07% | 99.5±0.66% |
| KMNpatch(1.1) - MN(2.2) | no | 99.97±0.04% | 68.78±20.03% |

Table 1: Results for the Dominoes experiment averaged across 4-runs. Numbers in bracket show mean-PD (dataset difficulty). Core-only accuracy indicates the model's reliance on core features. Models achieve high core-only accuracy when spurious features are harder than core features.

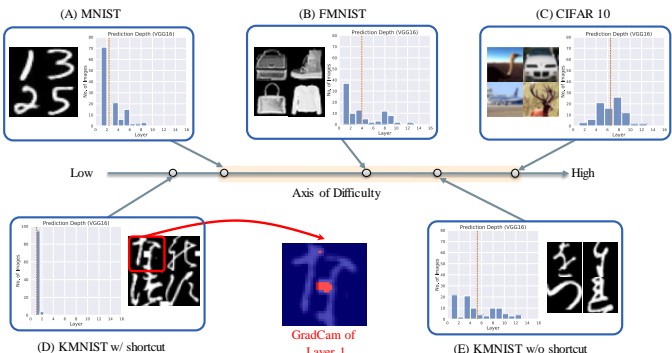

Figure 4: Top row shows three reference datasets at three different levels of difficulty and their corresponding prediction depth (PD) plots. The datasets are ordered based on their difficulty (measured using mean PD shown by dotted vertical lines). The bottom row shows the effect of the shortcut on the KMNIST dataset. The yellow region on the axis indicates the expected difficulty of classifying KMNIST. While the original KMNIST lies in the yellow region, the shortcut significantly reduces the task difficulty. The Grad-CAM shows that the model focuses on the spurious patch.

From Table 1, we observe that all datasets achieve a high test accuracy as expected. The core-only accuracy stays high ($> 98\%$) for datasets where the spurious feature is harder to learn than the core feature, indicating the model's high reliance on core features. When the spurious feature is easier than the core, the model learns to leverage them, and hence the core-only accuracy drops to nearly random chance ( 50%). This experiment demonstrates that a spurious feature that is harder than the core feature fails to act as a shortcut that the model can leverage to "cheat".

## 4.2 MONITORING INITIAL LAYERS CAN REVEAL SUSPICIOUS SHORTCUT LEARNING ACTIVITY

**Synthetic Shortcut on Toy Dataset:** To provide a proof of concept, we demonstrate our method on the Kuzushiji-MNIST (KMNIST) (Clanuwat et al., 2018) dataset comprising Japanese Hiragana characters. The dataset has ten classes and images of size $28 \times 28$, similar to MNIST. We insert a white patch (shortcut) at a particular location for each of the ten classes. The location of the patch is class-specific. We train two VGG16 models, one on the KMNIST with a patch shortcut ($\mathcal{M}_{sh}$) and another on the original KMNIST without the patch ($\mathcal{M}_{orig}$).

Fig-4 shows the prediction depth (PD) plots for this experiment. The vertical dashed lines show the mean PD for each plot. Intuitively KMNIST should be harder than MNIST but easier than CIFAR10. But introducing the white patch makes KMNIST easier than even MNIST for $\mathcal{M}_{sh}$ (see Fig - 4A & 4D). The white patch is a very easy feature, and hence the model only needs a single layer to detect it. The Grad-CAM maps for the layer-1 show that $\mathcal{M}_{sh}$ focuses mainly on the patch (see Fig-4D), and hence the test accuracy on the original KMNIST images is very low ( 8%). The PD plot for $\mathcal{M}_{orig}$ (see Fig-4E) is not as skewed toward lower depth as the plot for $\mathcal{M}_{sh}$. This is expected as $\mathcal{M}_{orig}$ is not looking at the shortcut and therefore utilizes more layers to make the prediction. The mean PD for $\mathcal{M}_{orig}$ suggests that the original KMNIST is harder than Fashion-MNIST but easier than CIFAR10, which is intuitive. $\mathcal{M}_{orig}$ also achieves a higher test accuracy ( 98%).

This experiment demonstrates how models that learn shortcuts ($\mathcal{M}_{sh}$) exhibit PD plots that are suspiciously skewed towards the initial layers. If the dataset is sufficiently difficult, the skewed PD plot should raise concerns. Visualization techniques like Grad-CAM can further aid our intuition about what shortcut the model is utilizing at any given layer.

**Semi-Synthetic Shortcut on Medical Datasets:** We follow the procedure by DeGrave et al. (2021) to create the ChestX-ray14/GitHub-COVID dataset. This dataset comprises Covid19 positive images from Github Covid repositories and negative images from ChestX-ray14 dataset (Wang et al., 2017b). In addition, we also create the Chex-MIMIC dataset following the procedure by Puli et al. (2022). This dataset comprises 90% images of Pneumonia from Chexpert (Irvin et al., 2019)

and 90% healthy images from MIMIC-CXR (Johnson et al., 2019). We train two DenseNet121 models, $\mathcal{M}_{covid}$ on the ChestX-ray14/GitHub-COVID dataset, and $\mathcal{M}_{chex}$ on the Chex-MIMIC dataset. We use DenseNet121, a common and standard architecture for medical image analysis. Images are resized to $512 \times 512$.

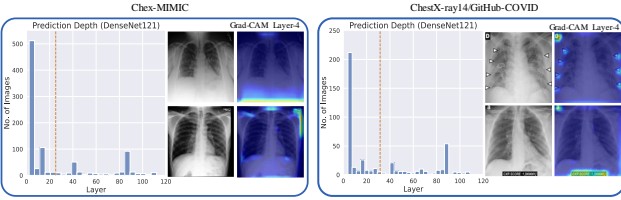

Figure 5: PD plots for two DenseNet-121 models trained on Chex-MIMIC and ChestX-ray14/GitHub-COVID datasets are shown in the figure, along with their corresponding Grad-CAM visualizations. Both PD plots exhibit a very high peak in the initial layers (1 to 4), indicating that the models use very easy features to make the predictions.

Fig-5 shows the PD plots for $\mathcal{M}_{chex}$ and $\mathcal{M}_{covid}$. Both the plots are highly skewed towards initial layers, similar to the KMNIST with patch shortcut in Fig-4D. This again indicates that the models are using very easy features to make the predictions, which is counterintuitive as the two tasks (pneumonia and covid19 detection) are hard tasks even for humans. Examining the Grad-CAM maps at layer-4 reveals that these models focus on irrelevant spurious features outside the lung region. This raises concern because both diseases are known to affect mainly the lungs. The reason for this suspicious behavior is that, in both these datasets, the healthy and diseased samples have been acquired from two different sources. This creates a shortcut because source-specific attributes or tokens are predictive of the disease and can be easily learned, as pointed out by DeGrave et al. (2021).

**Real Shortcut on Medical Dataset:** For this experiment, we use the NIH dataset (Wang et al., 2017a) which has the popular chest drain shortcut (for Pneumothorax detection) (Oakden-Rayner et al., 2020). Chest drains are used to treat positive Pneumothorax cases. The presence of a chest drain in the lung is, therefore, positively correlated with the presence of Pneumothorax and can be used by the deep learning model (Oakden-Rayner et al., 2020). Appendix-A.3 outlines the procedure we use to obtain chest drain annotations for the NIH dataset. We train a DenseNet121 model ($\mathcal{M}_{nih}$) for Pneumothorax detection on NIH images of size $128 \times 128$.

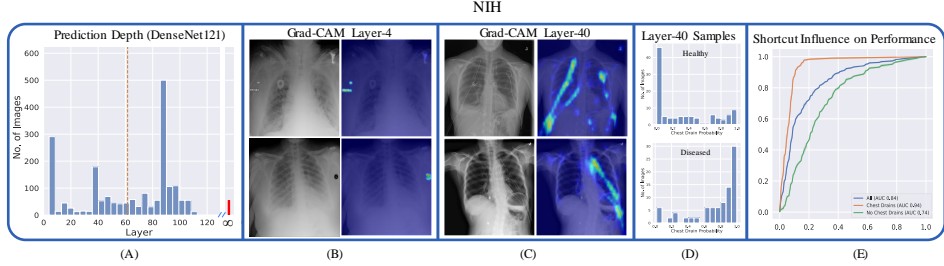

Figure 6: Shortcut learning on NIH dataset. (A) PD plot for DenseNet-121 trained on NIH shows three prominent peaks in layers-4,40,88. (B, C) Grad-CAM reveals that layer-4 uses irrelevant artifacts as shortcuts, and layer-40 uses the chest drain shortcut for classification. (D) Plotting chest drain (shortcut) probability for layer-40 samples reveals that the model segregates healthy patients from diseased ones based on the shortcut: most diseased patients have a chest drain, whereas most healthy patients do not. (E) The chest drain shortcut affects the AUC performance of the model. The X-axis (Y-axis) shows the false positive (true positive) rate.

Fig-6A shows the PD plot for $\mathcal{M}_{nih}$. We observe that the distribution is not as skewed as the plots in the previous experiments. This is because all the images come from a single dataset (NIH). But we see two suspicious peaks at layers - 4 & 40. Pneumothorax classification is challenging even for radiologists, and hence peaks at initial layers raise suspicion. The Grad-CAM maps in Figs -6B &

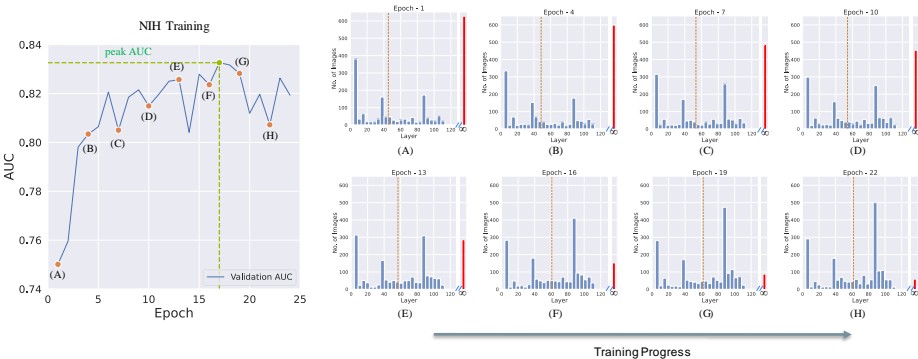

Figure 7: Evolution of PD plot across epochs shows the training dynamics of the DNN on the NIH dataset. The initial peaks (layers-4&40) are relatively stable throughout training, whereas the later peaks (layer-88) change with time. The initial layers learn the easy shortcuts, which can be detected early during the training. Samples with undefined PD (shown in red) take more time to converge and eventually accumulate in the later layers (layer 88 in this case).

6C reveal that layer-4 looks at irrelevant artifacts in the chest X-ray image, whereas layer-40 looks at chest drain in the image. The chest drain shortcut is much harder and is therefore detected at layer 40. To verify that layer-40 is looking for chest drains, we ran a tube detector on the images at layer-40 and found that most diseased patients had a chest drain inserted into their lungs, while most healthy patients did not have the chest drain (see Fig-6D). This provides evidence that layer-40 categorizes patients based on the chest drain shortcut. Appendix-A.3 provides details on how we train a tube detector. Fig-6E shows that the AUC performance is 0.94 when the diseased patients have a chest drain and 0.74 when they don't. In both cases, the set of healthy patients remains the same. This observation is consistent with the findings of Oakden-Rayner et al. (2020) and indicates that the model looks at chest drains to classify positive Pneumothorax cases.

The above experiments demonstrate how a peak located in the initial layers of the PD plot should raise suspicion, especially when the classification task is challenging. Visualization techniques like Grad-CAM can further aid our intuition and help identify the shortcuts being learned by the model. This approach works well even for realistic scenarios and challenging shortcuts (like chest drain for Pneumothorax classification), as shown above.

### 4.3 DETECTING SHORTCUTS EARLY

Fig-7 shows the evolution of the PD plot across epochs for $\mathcal{M}_{nih}$ (which is the model used in Fig-6). This visualization helps us observe the training dynamics of the various layers. The red bar in the PD plots shows the samples with undefined prediction depths.

These plots reveal several useful insights into the learning dynamics of the model. Firstly, we see three prominent peaks in epoch-1 at layers-4,40,88 (see Fig-7A). The magnitude of the initial peaks (like layers-4&40) remains nearly constant throughout the training. The peaks at layers-4&40 correspond to shortcuts, as discussed in the previous section. This indicates that easy shortcuts can often be identified early (epoch-1 in this case). Fig-8 shows the PD plots at epoch-2 for other datasets with shortcuts. It is clear from Fig-8 that the suspiciously high peak at the initial layer is visible in the *second epoch* itself. The Grad-CAM maps reveal that this layer looks at irrelevant artifacts in the dataset. This behavior is seen in all datasets shown in Fig-8.

Secondly, we also see that accuracy or AUC plots do not reveal shortcut learning patterns, and we need to monitor the training dynamics using suitable metrics (like PD) to detect this behavior. Thirdly, the red peak (undefined samples) decreases in magnitude with time, and we see a proportional increase in the layer-88 peak. This corroborates well with the observation that later layers take more time to converge (Rahaman et al., 2019; Mangalam & Prabhu, 2019; Zeiler & Fergus, 2014). Therefore, samples with higher PD are initially undefined and do not appear in the PD plot. Nonetheless, samples with lower PD show up very early during the training, and this helps us in the

Figure 8: Epoch-2 PD plots for various datasets with shortcuts. The high spurious peak in the initial layer is visible in all the datasets indicating that shortcuts can be detected early during the training.

early detection of shortcuts. Early detection can consequently help develop intervention schemes that fix the shortcut early.

## 4.4 PREDICTION DEPTH $\approx \mathcal{V}$-USABLE INFORMATION

Table-2 measures the influence of shortcuts on NIH and KMNIST using PD and PVI metrics. All diseased patients in the "NIH w/ Shortcut" dataset have a chest drain, whereas all diseased patients in the "NIH w/o Shortcut" dataset have no chest drain. The set of healthy patients is common for the two datasets. The KMNIST datasets are the same as those used in Section-4.2. We use VGG16 for KMNIST and DenseNet121 for NIH. Other training details remain the same as in Section-4.2.

| Dataset | mean PD | $-H_{\mathcal{V}_{cnn}}(Y \mid X)$ |
|---|---|---|
| NIH w/ Shortcut | 53.43 | -0.1171 |
| NIH w/o Shortcut | 75.33 | -0.2321 |
| KMNIST w/ Shortcut | 1.06 | -0.0024 |
| KMNIST w/o Shortcut | 5.25 | -0.0585 |

Table 2: Effect of Shortcuts on Prediction Depth and the negative conditional $\mathcal{V}$-entropy (- $H_{\mathcal{V}_{cnn}}(Y \mid X)$). The label marginal distributions for NIH and KMNIST are the same with or without shortcut, and thus the negative conditional $\mathcal{V}$-entropy is proportional to $\mathcal{V}$-information.

Table-2 shows that datasets with shortcuts ($D_s$) have smaller mean PD values than their counterparts without shortcuts ($D_i$). Proposition-1 (see Section-3, Appendix-A.1) shows that a sufficient gap between the mean PDs of $D_s$ and $D_i$ causes the $\mathcal{V}$-Information of $D_s$ to be greater than $D_i$. Table-2 confirms this in a medical-imaging dataset with a real shortcut, and we see that the mean "usable information" increases when there is a shortcut. This implies that the model learns shortcut features as they have more usable information than the core features. We investigate this relationship between PD and $\mathcal{V}$-information in more detail in Appendix-A.8. Ethayarajh et al. (2021) also show that $\mathcal{V}$-information is positively correlated with test accuracy. This explains the significant change in AUC observed in Fig-6E. Proposition 1 bridges the gap between the notions of PD and $\mathcal{V}$-usable information. This connection between $\mathcal{V}$-information and PD indicates that monitoring early training dynamics using PD not only helps detect shortcut learning but also bears insights into the dataset's difficulty (in information-theoretic terms) for a given model class.

## 5 CONCLUSION

In this paper, we study shortcut learning by monitoring the early training dynamics of DNNs. We emphasize that not all spurious correlations are shortcuts and empirically show that models suffer from shortcut learning only when the spurious features are easier than the core features. We hypothesize that *"Potential shortcuts can be found by monitoring the easy features learned by the initial layers of a DNN early during the training."* and validate this hypothesis on real medical imaging data. We empirically demonstrate that shortcuts are learned quite early during the training and that one can detect shortcut learning by monitoring the early training of DNNs using suitable instance difficulty metrics like Prediction Depth (PD). Further, we show a theoretical connection between PD and $\mathcal{V}$-information to support our empirical results. Datasets with shortcuts have more $\mathcal{V}$-information causing the model to learn the shortcut. To conclude, relying only on accuracy plots is insufficient, and we recommend monitoring the DNN training dynamics using additional metrics like PD for the early detection of shortcuts.

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

# A APPENDIX

## A.1 PROOF OF PROPOSITION-1

**Proposition 1.** *Given two datasets, $D_s$ with shortcuts and $D_i$ without shortcuts, we assume the following:*

1. *(Well-Trained Model Assumption) The part of the network from any representation to the label is one of the functions that compute $\mathcal{V}$-information.*

2. *(Function Class Complexity Assumption) Assume that there exists a $K \in \{1, N\}$ such that $V_{cnn}$ of depth $N - K$ is deep enough to be a strictly larger function class than $V_{knn}$ with a fixed neighbor size (29 in this paper). Assume that this $V_{knn}$ is a larger function class than a linear function.*

3. *(Controlled Confidence Growth Assumption) For both datasets $D \in \{D_s, D_i\}$, assume that the for all $k \in \{1, \cdots, N\}$,*

$$\tau \leq \mathcal{I}_{\mathcal{V}_{knn}}^D(\phi_k) - \mathcal{I}_{\mathcal{V}_{knn}}^D(\phi_{k-1}) \leq \epsilon$$

4. *(Prediction Depth Separation Assumption) Let $L$ be an integer such that, $L \leq K$ and $L < K + (N - K)\frac{\epsilon}{\tau} - \psi \max_y \left(-\log p(Y = y)\right).$*

*Let there exist a gap in prediction depths of samples in $D_s$ and $D_i$ : $\psi \in (0, 0.5)$ such that $1 - \psi$ fraction of $D_s$ has prediction depth $\leq L$ and $1 - \psi$ fraction of $D_i$ has prediction depth $> K$.*

*Then, for a model class of $N$-layer CNNs, we show that the $\mathcal{V}_{cnn}$-information for $D_s$ is greater than $\mathcal{V}_{cnn}$-information for $D_i$:*

$$\mathcal{I}_{\mathcal{V}_{cnn}}^{D_s}(X \to Y) \geq \mathcal{I}_{\mathcal{V}_{cnn}}^{D_i}(X \to Y)$$

*Proof.* We proceed in two parts: first, we lower bound $\mathcal{V}_{cnn}$-information for $D_s$, and then we upper bound $\mathcal{V}_{cnn}$ for $D_i$.

Assumption 3 implies:

$$(B - A)\tau \leq \sum_{k=A}^{B} \tau \leq \mathcal{I}_{\mathcal{V}_{knn}}^D(\phi_k) - \mathcal{I}_{\mathcal{V}_{knn}}^D(\phi_{k-1}) \leq (B - A)\epsilon$$

**PD - PVI connection.** Note that by definition, when the prediction depth is $k$ for a sample $X$, then $PVI_{knn}(\phi_k(X)) \geq \delta$ but $PVI_{knn}(\phi_{k-1}(X)) < \delta$. This follows from how we compute PD (see Section-3 in main paper, and Appendix-A.4).

**Lower bounding $\mathcal{I}_{\mathcal{V}_{cnn}}^{D_s}$**

$$
\begin{aligned}
\mathcal{I}_{\mathcal{V}_{cnn}}^{D_s} &= \mathcal{I}_{\mathcal{V}_{cnn \text{ of depth } N-K}}^{D_s}(\phi_K) && \text{\{Assumption-1\}} \\
&\geq \mathcal{I}_{\mathcal{V}_{knn}}^{D_s}(\phi_K) && \text{\{Assumption-2\}} \\
&= \mathcal{I}_{\mathcal{V}_{knn}}^{D_s}(\phi_L) + \sum_{k=L+1}^{K} \mathcal{I}_{\mathcal{V}_{knn}}^D(\phi_k) - \mathcal{I}_{\mathcal{V}_{knn}}^D(\phi_{k-1}) && \text{\{Telescoping Sum\}} \\
&\geq \mathcal{I}_{\mathcal{V}_{knn}}^{D_s}(\phi_L) + (K - L)\tau && \text{\{Assumption-3\}} \\
&\geq \psi \min_{X,Y \in D_s, pd>=L} PVI_{knn}(X \to Y) \\
&\quad + (1 - \psi) \min_{X,Y \in D_s, pd<L} PVI_{knn}(X \to Y) + (K - L)\tau && \text{\{Prediction Depth Separation\}} \\
&\geq 0 * \psi + \delta * (1 - \psi) + (K - L)\tau && \text{\{Prediction Depth Separation\}}
\end{aligned}
$$

**Upper bounding $\mathcal{I}_{\mathcal{V}_{cnn}}^{D_i}$**

$$\mathcal{I}_{\mathcal{V}_{cnn}}^{D_i} \leq \mathcal{I}_{\mathcal{V}_{knn}}^{D_i}(\phi_N) \qquad \text{\{Assumption-2\}}$$

$$= \mathcal{I}_{\mathcal{V}_{knn}}^{D}(\phi_K) + \sum_{k=K+1}^{N} \mathcal{I}_{\mathcal{V}_{knn}}^{D}(\phi_k) - \mathcal{I}_{\mathcal{V}_{knn}}^{D}(\phi_{k-1}) \qquad \text{\{Telescoping Sum\}}$$

$$\leq \mathcal{I}_{\mathcal{V}_{knn}}^{D}(\phi_K) + (N-K)\epsilon \qquad \text{\{Assumption-3\}}$$

$$\leq (N-K)\epsilon + \psi \max_{X,Y \in D_i, pd(X) \leq K} PVI_{\mathcal{V}_{knn}}^{D}(\phi_K(X) \to Y)$$

$$+ (1-\psi) \max_{X,Y \in D_i, pd(X) > K} PVI_{\mathcal{V}_{knn}}^{D}(\phi_K(X) \to Y) \qquad \text{\{Prediction Depth Separation\}}$$

$$\leq (N-K)\epsilon + \psi \max_{y} (-\log p(Y = y))$$

$$+ (1-\psi) \max_{X,Y \in D_i, pd(X) > K} PVI_{\mathcal{V}_{knn}}^{D}(\phi_K(X) \to Y) \qquad \text{\{PVI} \leq -\log p(Y = y)\text{\}}$$

$$\leq (N-K)\epsilon + \psi \max_{y} (-\log p(Y = y)) + (1-\psi)\delta \qquad \text{\{PD-PVI connection for pd} > K\text{\}}$$

The proof follows by comparing the lower bound on $\mathcal{I}_{\mathcal{V}_{cnn}}^{D_s}$ and the upper bound on $\mathcal{I}_{\mathcal{V}_{cnn}}^{D_i}$. Intuitively what this means is that when there is a sufficiently large gap in the mean PD between $D_s$ and $D_i$, then the $\mathcal{V}$-information of $D_s$ exceeds the $\mathcal{V}$-information of $D_i$, which is why the model prefers learning the shortcut rather than using the core features for the task.

$\square$

## A.2 GRAD-CAM VISUALIZATION

PD plots help us understand the layers of the model which are actively used for classifying different images. To further aid our intuition, we visualize the Grad-CAM outputs for an arbitrary layer $k$ of the model by attaching a soft-KNN head. Let $g_{knn}$ denote the soft and differentiable version of k-NN. We compute $g_{knn}$ as follows:

$$g_{knn}(\phi_q^k; \phi_{i \in \{1,2,...m\}}^k) = \frac{\sum_{j \in \mathcal{N}(\phi_q^k, 1)} \exp^{-\|\phi_q^k - \phi_j^k\|/s}}{\sum_{j \in \mathcal{N}(\phi_q^k, :)} \exp^{-\|\phi_q^k - \phi_j^k\|/s}}$$

This function makes the KNN differentiable and can be used to compute Grad-CAM (Selvaraju et al., 2017). We use the $\mathcal{L}_1$ norm for all distance computations. $\phi_q^k$ corresponds to feature at layer-$k$ for query image $x_q$. Let $\phi_{i \in \{1,2,...m\}}^k$ be the training data for KNN. Let $\mathcal{N}$ denote the neighborhood function. $\mathcal{N}(\phi_q^k, :)$ returns the indices of K-nearest neighbors for $\phi_q^k$. $\mathcal{N}(\phi_q^k, 1)$ returns indices of images with positive label ($y = 1$) from the set of K-nearest neighbors for $\phi_q^k$. $s$ is the median for the set of $\mathcal{L}_1$ norms $\{\|\phi_q^k - \phi_j^k\|\}$ for $j \in \mathcal{N}(\phi_q^k, :)$.

## A.3 CHEST DRAIN ANNOTATIONS FOR NIH DATASET

To reproduce the results by Oakden-Rayner et al. (2020), we need chest drain annotations for the NIH dataset (Wang et al., 2017a), which is not natively provided. To do this, we use the MIMIC-CXR dataset (Johnson et al., 2019), which has rich meta-data information available in the form of radiology reports. We collaborate with radiologists to identify terms related to Pneumothorax from the MIMIC-CXR reports. These include pigtail catheters, pleural tubes, chest tubes, thoracostomy tubes, etc. We collect chest drain annotations for MIMIC-CXR by parsing the reports for these terms using the RadGraph NLP pipeline (Jain et al., 2021). We train a DenseNet121 model to detect chest drains relevant to Pneumothorax using these annotations. Finally, we run this trained model on the NIH dataset to obtain the needed chest drain annotations. We use these annotations to obtain the results shown in Figs - 6D and 6E. Fig-6E closely reproduces the results obtained by Oakden-Rayner et al. (2020).

## A.4 Notion of Undefined Prediction Depth

Section-3 shows how we compute PD in our experiments. While fully trained models give valid PD values, our application requires working with arbitrary deep-learning models that are not necessarily fully trained. We, therefore, introduce the notion of undefined PD by treating $k$-NN predictions close to 0.5 (for a binary classification setting) as invalid. We define a $\delta$ such that $|f_{knn}(x) - 0.5| < \delta$ implies an invalid $k$-NN output. We use $\delta = 0.1$ and $k = 29$ in our experiments. If any $k$-NN predictions for the last three layers are invalid, we treat the PD of the input image to be undefined. To work with high-resolution images (like $512 \times 512$), we downsample the spatial resolution of all training embeddings to $8 \times 8$ before using the $k$-NN classifiers on the intermediate layers. We empirically see that our results are insensitive to $k$ in the range $[5, 30]$.

## A.5 PD Plots at Epoch-0

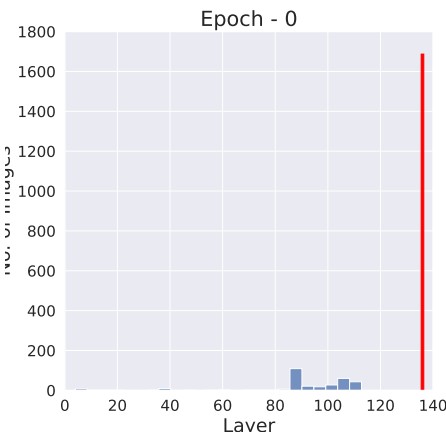

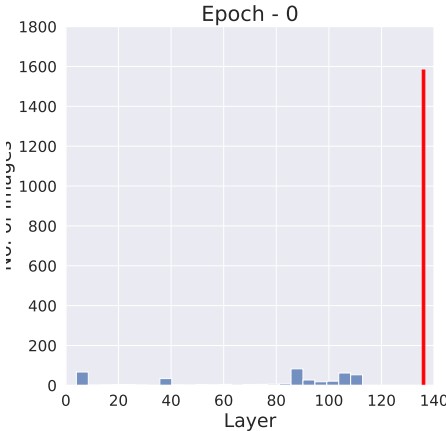

Figure 9: Pneumothorax Detection in NIH

Figure 10: Age Detection in NIH

We show epoch-0 PD plots for various tasks on the NIH dataset (see Figures-9&10). We expect most samples to be undefined at epoch 0 (shown by the red bar), and we see some additional noisy patterns in the plot. The model is not trained during initialization, and hence the weights and features are random vectors carrying little information. Most of the instance difficulty metrics are ill-defined until the model is fully trained, and therefore we find this uninterpretable behavior in PD plots at epoch-0. Although most of the instance difficulty metrics fail to give meaningful information at the start of training Ethayarajh et al. (2021); Agarwal et al. (2022), we empirically observe that PD plots can capture the distribution of the relatively easy samples in the training dataset even during early stages of training. This favors our hypothesis, which relies on monitoring easy samples in early layers to detect shortcut learning. We, therefore, use PD over other metrics to analyze the early training dynamics of the model.

## A.6 Code Reproducibility

Please find the code here: `https://github.com/anonymCloud/ICLR2023_Shortcut_Learning_Through_Training_Dynamics`

## A.7 Vision Experiments

We use the *NICO++* (Non-I.I.D. Image dataset with Contexts) dataset Zhang et al. (2022) to create multiple spurious datasets (Cow vs. Bird; Dog vs. Lizard; Flower vs. Lizard) such that the context/background is spuriously correlated with the target. NICO++ is a Non-I.I.D image dataset that uses context to differentiate between the test and train distributions. This forms an ideal setup to investigate what spurious correlations the model learns during training. We follow the procedure outlined by Puli et al. (2022) to create datasets with spurious correlations (90% prevalence) in the

training data, and the test data has the relationship between spurious attributes and the true labels flipped. This is similar to the Chex-MIMIC dataset illustrated in section-4.2. We test our hypothesis using ResNet-18 and VGG16. We train our models for 30 epochs using an Adam optimizer and a base learning rate of 0.01. We choose the best checkpoint using early stopping.

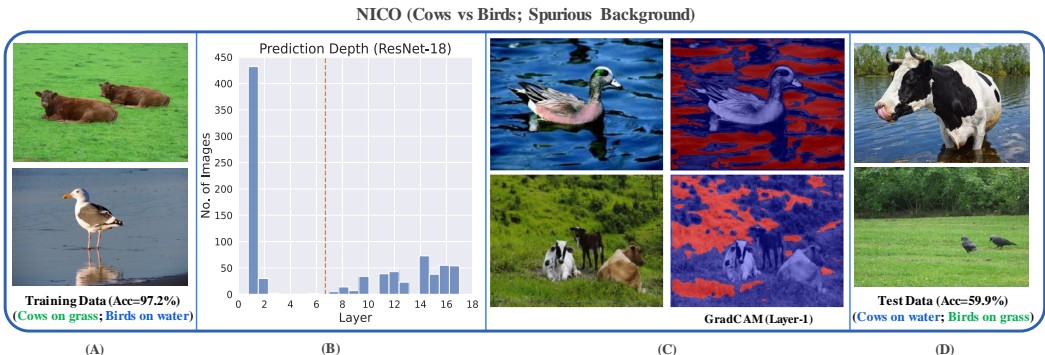

Figure 11: Cow vs. Birds classification on NICO++ dataset. (A) Training data contains images of cows on grass and birds on water (correlation strength=0.9). The model achieves 97.2% training accuracy. (B) PD plot for ResNet-18 reveals a spurious peak at layer-1, indicating the model's heavy reliance on very simple (potentially spurious) features. (C) GradCAM plots for layer 1 reveal that the model mainly relies on the spurious background to make its predictions. (D) Consequently, the model achieves a test accuracy of only 59.9% on test data where the spurious correlation is flipped (i.e., cows (birds) are found on water (grass)).

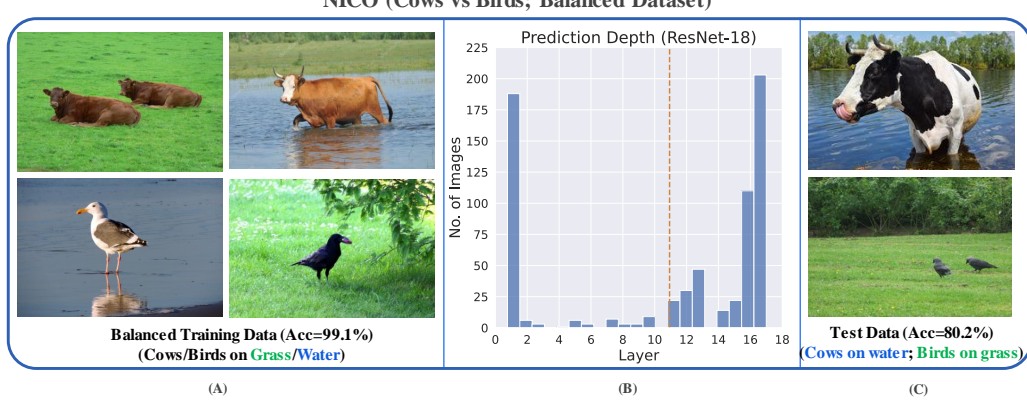

Figure 12: Balanced dataset for Cow vs. Birds classification task on NICO++ dataset. (A) The training dataset contains a balanced distribution of cows and birds found on water and grass (each group has an equal number of images). (B) The balanced dataset shifts the PD plot towards the later layers (compared to Fig-11B, indicating that the model relies on both simple and complex features for the prediction task. (C) This consequently results in an improved test accuracy of 80.2% (as compared to 59.9% in Fig-11D for the spurious dataset).

Figures-11,13,14 show PD plots and train/test accuracies for models that learn the spurious background feature present in the NICO++ dataset. While all models achieve $> 85\%$ training accuracy, they have poor accuracies ( 50%) on the test data where the spurious correlation is flipped. This can be seen simply by observing the PD plots for the model on the training data. The plots are skewed towards the initial layers indicating that the model relies heavily on very simple (potentially spurious) features for the task. GradCAM maps also confirm that the model often focuses on the background context rather than the foreground object of interest.

We further observe in Fig-12 that balancing the training data (to remove the spurious correlation) results in a model with improved test accuracy (80.2%) as expected. This is also reflected in the PD plot (Fig-12B), where we see that the distribution of the peaks, as well as the mean PD, shift

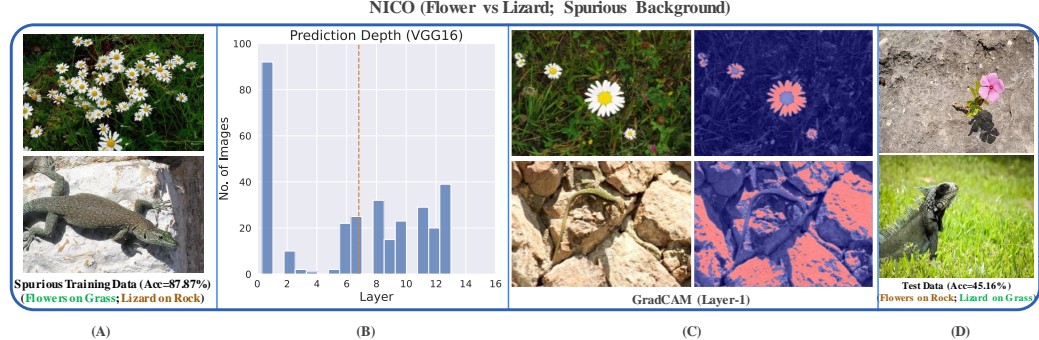

**Figure 13:** Dog vs. Lizard classification with a spurious background feature on NICO++ dataset. (A) Training data contains images of outdoor dogs and lizards on rock (correlation strength=0.9). The spurious background color/texture reveals the foreground object. The model achieves 87.2% training accuracy. (B) PD plot for ResNet-18 reveals a spurious peak at layer-1, indicating the model's reliance on simple (potentially spurious) features. (C) The low test accuracy confirms this (63.9%). The test data has the spurious correlation flipped (i.e., images contain dogs on rock and lizards found outdoors.)

**Figure 14:** Flower vs. Lizard classification with a spurious background feature on NICO++ dataset. (A) Training data contains images of flowers on grass and lizards on rock (correlation strength=0.9). The spurious background reveals the target class. The model achieves 87.87% training accuracy. (B) PD plot for ResNet-18 reveals a spurious peak at layer-1. (C) GradCAM plots for layer 1 confirm that the model uses the rock texture in the background to detect lizards, whereas it uses simple (non-spurious) color features to detect the flowers in the image. (D) Due to partial reliance on spurious features, the model obtains low test accuracy (45.16%) on the test data with the spurious correlation flipped (i.e., images contain flowers on rock and lizards on grass.)

proportionately towards the later layers, indicating that the model is now using a combination of both easy and complex features for the prediction task.

By monitoring PD plots during training and using suitable visualization techniques, we show that one can obtain useful insights about the spurious correlations that the model may be learning. This can also help the user make an educated guess about the generalization behavior of the model on various test datasets with different distributions.

## A.8 EMPIRICAL RELATIONSHIP: PD VS $\mathcal{V}$-INFORMATION

In Section-4.4 we explore the relationship between PD and $\mathcal{V}$-information. To empirically confirm these results, we further investigate this relationship on four additional datasets: KMNIST, FMNIST,

SVHN, and CIFAR10. We train a VGG16 model on these datasets for ten epochs using an Adam optimizer and a base learning rate of 0.01. We use a bar plot to show the correlation between PD and $\mathcal{V}$-entropy. We group PD into intervals of size four and compute the mean $\mathcal{V}$-entropy for samples lying in this PD interval.

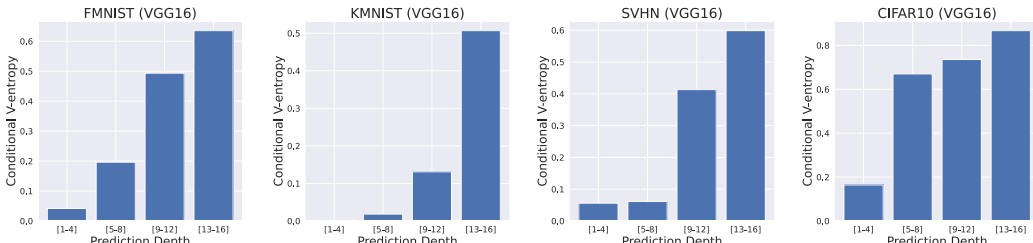

Figure 15: The bar plots show a positive correlation between PD and Conditional $\mathcal{V}$-entropy. Samples with higher PD also have a higher $\mathcal{V}$-entropy resulting in lower usable information for models like VGG16.

In Section-4.4, we find that PD is positively correlated with $\mathcal{V}$-information, and the results shown in Fig-15 further confirm this observation. Instance difficulty increases with PD, and the usable information decreases with an increase in $\mathcal{V}$-entropy. It is, therefore, clear from Fig-15 that samples with a higher difficulty (PD value) have lower usable information, which is not only intuitive but also provides empirical support to Proposition-1 in Appendix-A.1.

