# OpenReview forum: "Shortcut Learning Through the Lens of Early Training Dynamics"
_ICLR.cc/2023/Conference — Submitted to ICLR 2023_

### Official Review · Reviewer_5RsT · 2022-10-24

**Confidence:** 4
**Correctness:** 2
**Technical Novelty And Significance:** 2
**Empirical Novelty And Significance:** 2
**Recommendation:** 3

**Clarity, Quality, Novelty And Reproducibility:**

**Clarity:** Mosty clear. While the concept of undefined prediction depth is explained in appendix A.4, it might help to make it more clear in the main paper as well.

**Quality:** From a technical perspective, the experiments look mostly sensible, however there are two major concerns with the approach:
1. Shortcut definition. The authors introduce a distinction between shortcuts and spurious features, where shortcuts are only those features that are easy to learn. This definition contrasts with the existing definition of "Shortcut learning in deep neural networks" in which shortcuts are defined independently of "easyness". The approach of relying on "easy to learn" comes with a number of problems: now, one can no longer ask whether a dataset has a shortcut since the "easy to learn" definition is inherently tied to a specific model - while some features are easy to learn for some models, the same features can be hard / impossible to learn for others. (This is a consequence of the no free lunch theorem: averaged across all datasets the performance of any two models is equal.) Furthermore, the authors fall short of providing a convincing definition of "easy to learn", instead relying on a proxy, the prediction difficulty method. If shortcuts are defined as easy to learn, then the observation that shortcuts are learned first is trivial. It doesn't help that the authors aren't always using their own definition either: "First, we consider two datasets, one where the shortcut is easier than the core feature and another where the shortcut is harder." -> according to their definition, shortcuts are always easy to learn, so the existence of shortcuts which are harder than other features is inconsistent with the definition that the authors themselves advocate for. At points, the authors revert to visual intuition for defining "easy to learn", e.g. when stating that it is "visually intuitive that MNIST is easier than FMNIST, which is easier than CIFAR10". In summary, this goes to show that there are major problems when defining shortcuts relative to "easy to learn", especially without applying the necessary rigor in doing so consistently and with a convincing definition.
2. Circular argumentation: shortcuts are _defined_ as easy to learn. The authors then "empirically show that models suffer from shortcut learning only when the spurious features are easier than the core features." -> this is pretty much a natural consequence of the definition: if shortcuts are defined as easy to learn by a model, then it is no surprise that the easy to learn features are learned easily.

**Reproducibility:** Poor. The main proof is in the appendix. No code has been submitted, which is less than ideal from a reproducibility perspective. The appendix contains insufficient details on model training - e.g. it is simply stated that "We train a DenseNet121 model to detect chest drains relevant to Pneumothorax using these annotations." without providing any further details on the training (e.g. hyperparameters). The same is true for the models used in the main paper. I would encourage the authors to either submit code, or state why this is not possible; and additionally provide enough details that the findings can be reproduced without any trial and error. Typically, this includes details on hard- and software (including version numbers), all hyperparameters, links to existing training scripts that were used, etc.

**Novelty:** Limited. The paper starts with very simplistic settings (binary classification) and goes on to more realistic datasets. However, in all cases the shortcuts are known beforehand or even introduced by the authors themselves, thus it is unclear whether the method helps in finding novel shortcuts that were not known in advance. Finding a needle in a haystack is much easier if one knows the location beforehand. Furthermore, there is an open question regarding the usefulness of the model - if GRAD-CAM can show the shortcut, why do we need an additional method on top? Furthermore, the proposition (connection to V-information) seems close to obvious if I'm not mistaken: if one introduces more information, there's more information for the model to pick up on.


**Strength And Weaknesses:**

**Strenghts:**
- paper connects a few different ideas - prediction depth / example difficulty, early training dynamics, V-information
- the question of which features models learn, and whether those are intended or shortcut features, is an important one

**Weaknesses:**
- inconsistent and problematic shortcut definition
- circular argumentation
- poor reproducibility
- limited novelty: paper never provides evidence of finding shortcuts that were not known in advance; theoretical proof - if my understanding is correct - is not surprising/unexpected at all (if one introduces more information, then a model can use more information)


**Summary Of The Paper:**

The paper "SHORTCUT LEARNING THROUGH THE LENS OF EARLY TRAINING DYNAMICS" introduces a distinction between spurious correlations and shortcuts, where shortcuts are defined as "easy to learn" by some model. Through a few toy experiments as well as real-world datasets (e.g. chest X-ray), the authors then show that models preferably learn easy features (as identified via prediction depth, an existing technique that trains linear classifiers at each layer to identify the earliest layer after which an example is consistently correctly classified by all subsequent layers). The authors connect their findings to the existing concept of V-usable information and argue that looking at accuracy alone is insufficient to identify shortcuts. The main finding is summarized in this sentence from the paper: "experiments demonstrate how a peak located in the initial layers of the PD plot [prediction depth] should raise suspicion, especially when the classification task is challenging. Visualization techniques like Grad-CAM can further aid our intuition and help identify the shortcuts being learned by the model".


**Summary Of The Review:**

The paper introduces a new definition of shortcuts which is tied to the notion of "easy to learn", without providing convincing arguments / a clear model-independent definition of easy to learn features. The authors show that the existing method of "prediction depth" can be useful in identifying potential shortcuts. While the paper is mostly clearly written, there are major problems with the definition of shortcuts (inconsistency), circular argumentation, poor reproducibility and limited novelty.

---

> ### Author Response · Authors · 2022-11-16
> **Summary Response (Part: 1/3)**
>
> We thank the reviewer for the detailed comments. We hope that our point-by-point reply will help address the issues. If not, we are looking forward to discussing it with you.
>
> **In summary:**
>
> - Shortcut definition is not only data dependent but also depends on the model. That is one of the main points of the paper.
> - The definition of “easy to learn” is not circular. Sorry for the confusion. We have two separate arguments in the paper: (1) When we have access to ***reference PD plots*** (see Fig-4 of the latest manuscript), one need not have apriori knowledge about the spurious feature $s$. (2) If reference PD plots aren’t available, one can still use our method to detect suspicious activity by monitoring the early training dynamics using PD plots. Further intuition can be obtained by visualizing what the model learns in early layers; this requires a human-in-the-loop for inspection.
> - To define task difficulty, we looked into two proxies, PD and $\mathcal{V}$-information. We theoretically and empirically show that these two are related. We need a proxy for task difficulty because it depends not only on the data but also on the model.
> - We provide a link to the anonymous Github repository in the paper (see Appendix-A.6), and other details (like python packages and version numbers) are also included to reproduce our results easily.

---

> > ### Comment · Reviewer_5RsT · 2022-11-25
> > **Reviewer response to rebuttal**
> >
> >
> > ### 1. shortcut definition: now definition is given; however definition seems problematic
> > I appreciate that the authors now provide a mathematical definition of their understanding of shortcuts, which resolves some ambiguities.
> > However, it looks like the definition is problematic (please correct me if I'm mistaken). According to definition 3, spurious feature s is a potential shortcut for model M if and only if the difficulty of predicting X -> y is higher (=more difficult) than predicting X -> s for the training distribution $P_{tr}$. If the spurious feature s is perfectly correlated with the true label y in the training distribution (i.e. something one could consider a perfect shortcut), then the difficulty of predicting X -> y is identical to predicting X -> s, which according to the given definition would not be considered a shortcut.
> >
> > Furthermore, it would be great if the authors could clarify whether $P_{te}$ is an i.i.d. test set, an o.o.d. test set, or just any test set without making the iid/ood distinction?
> >
> > ### 2. circular argumentation: concerns remain
> > I don't think the definition resolves the circular argumentation problematic.
> > The author's definition given in the rebuttal defines potential shortcuts as spurious features that are easier to learn than the core features.
> > The paper's summary states that they "empirically show that models suffer from shortcut learning only when the spurious features are easier than the core features". This is circular - if shortcuts are defined precisely as spurious features that are easier to learn than the core features, then of course models suffer from shortcut learning only when the spurious features are easier than the core features - by definition, not by experiment.
> >
> > ### 3. poor reproducibility: concerns resolved - thanks
> > I would like to thank the authors for providing more details on training details and for submitting their code. My concerns around reproducibility have been resolved. (Apologies for the unclear formulation regarding "The main proof is in the appendix." in my original review, this comment was meant to highlight the fact that the theoretical results can be followed by looking at the main proof in the appendix, which is totally fine)
> >
> > ### 4. limited novelty / knowing shortcuts beforehand: method does not now shortcuts; method users do
> > I think the authors misunderstood my comment; apologies for not being more clear. I fully understand that the proposed method does not have a priori access / information regarding shortcuts - however, the authors do. In all of the used datasets, the shortcuts that were "discovered" were known before to the authors, either since they introduced the shortcuts themselves or because they used datasets for which the shortcut was already known from prior literature. Therefore, it remains an open question whether the method can indeed help to find/discover a shortcut that was not known to the user of the method in advance. (I think it is not unlikely that it will help, but showing this would be better.)
> >
> > ### Summary:
> > In the light of the author's efforts regarding greater clarity (providing a shortcut definition) and improved reproducibility, I am increasing my score from 1 -> 3. That said, important concerns remain; most importantly regarding a potentially problematic definition and the circular argumentation (conflating definition with empirical results).
> >
> > (Note that I will be attending NeurIPS next week and will thus likely not be able to respond quickly; however I will make sure to check back afterwards.)

---

> > > ### Author Response · Authors · 2022-11-30
> > > **Clarifying Shortcut Definition, Circular Argumentation, Detecting Unknown Shortcuts (Response-2)**
> > >
> > > We thank the reviewer for the continuous engagement with us, despite traveling; we appreciate it:
> > >
> > > > **If the spurious feature s is perfectly correlated with the true label y in the training distribution (i.e. something one could consider a perfect shortcut), then the difficulty of predicting X \rightarrow y is identical to predicting X \rightarrow s, which according to the given definition would not be considered a shortcut.**
> > >
> > > A perfect correlation between $s$ and $y$ does not imply that the difficulty of predicting $X \rightarrow y$ is identical to predicting $X \rightarrow s$ since $\Psi(\cdot)$ is not the same as accuracy. However, we realize why our definition caused the confusion. We assume that $\Psi(X \rightarrow s)$ is given or computed from another dataset that is not fully correlated with y. If the only available data is that s and y are fully correlated, then we can see how the difficulty level is the same.  For example, in Table 1, both spurious and core features correlate perfectly with the true label. But we still see they have very different difficulty levels measured using PD. In the CIF-FMN dataset, CIFAR10 and FMNIST have mean-PD difficulties of 6.8 and 3.9, respectively.
> > >
> > > > **Furthermore, it would be great if the authors could clarify whether Pte is an i.i.d. test set, an o.o.d. test set, or just any test set without making the iid/ood distinction?**
> > >
> > > The causal graphs for $P_{te}$ and $P_{tr}$ are given in Fig 1. Samples from $P_{te}$ ($P_{tr}$) are iid samples from the graph on the right (left).
> > >
> > > > **Circular Argumentation**
> > >
> > > Our paper summary simply states that the model learns only those spurious features that are easier than the core features. We think the phrase “shortcut learning” in the summary statement is causing you confusion.
> > >
> > > Let's consider two kinds of spurious features, $s_1$ and $s_2$, for the main core task $M$: $X \rightarrow y$. At the *hindsight*, consider that $s_1$ and $s_2$ are *given* as labels, and one can construct discriminative models for $M_1$: $X \rightarrow s_1$ and $M_2$: $X \rightarrow s_2$. Now, consider both $M_1$ and $M_2$ work extremely well; e.g., the classifier $M_1$ and  $M_2$ obtain perfect classification and such that $\Psi_{M_1}(X \rightarrow s_1)<\Psi_{M}(X \rightarrow y)<\Psi_{M_2}(X \rightarrow s_2)$. Our *empirical observation (not theoretical results)* states that $s_1$ is problematic and $s_2$ is not. If we had stated that "when $s_2$ is not learnable hence it is not problematic", that would have been circular, but that is not what we are saying. We are saying that $s_2$ is learnable and relative to $s_1$ and $y$ is a harder task, hence not problematic.
> > >
> > > We respectfully disagree that we are conflating empirical and theory. Our empirical observation is based on PD. We theoretically relate PD to $\mathcal{V}$-information about task difficulty (we believe for the first time). Our theory does **NOT** prove that $s_1$ is problematic and $s_2$ is not; that requires new theoretical tools to prove that the basin of a "good solution" is not reachable to the learning dynamic when $s_1$ is present. We do not state that anywhere in the paper.
> > >
> > > > **Detecting unknown shortcuts**
> > >
> > > Detecting unknown shortcuts requires a human-in-the-loop to confirm/disprove that a feature/pattern is a shortcut. PD plot can hint toward such an issue, but humans are still needed to confirm this. That requires a falsifiable explainability method to visualize and show the results to a human observer. Designing such an experiment requires a human observer experiment, and it would be challenging to evaluate due lack of consensus about explainability methods. For that reason, we focused on cases where shortcuts are known to make evaluation more straightforward. However, the following is our anecdotal observation:
> > >
> > > *In the experiment in section 4.2 (Fig 5), we knew from the literature that a shortcut exists [Puli 22, DeGrave 21], but it was unclear what the shortcut is. Members of our lab have substantial experience with medical images. After visualizing the results, we noticed that one of the shortcuts was the cropping of samples.*
> > >
> > > **References**
> > > * DeGrave, Alex J., Joseph D. Janizek, and Su-In Lee. "AI for radiographic COVID-19 detection selects shortcuts over signal." Nature Machine Intelligence 3.7 (2021): 610-619.
> > > * Aahlad Manas Puli, Lily H Zhang, Eric Karl Oermann, and Rajesh Ranganath. Out-of-distribution generalization in the presence of nuisance-induced spurious correlations. In International Con- ference on Learning Representations, 2022.

---

> > > > ### Comment · Reviewer_5RsT · 2022-12-09
> > > > **reviewer response**
> > > >
> > > > Thanks for the response. Regarding the shortcut definition, it seems like we agree this definition excludes dataset where a perfectly correlated predictor is introduced (and thus the difficulty of predicting the true label can be identical to predicting the spurious feature). Since this is a setting that occurs in many commonly used shortcut datasets such as e.g. ShiftMNIST or any other dataset where an artificial perfectly correlated shortcut is introduced, I would recommend making this more clear in the paper to avoid confusing readers. Many people investigate shortcut learning in toy settings where perfect predictivity exists, e.g. by a class-predictive white pixel, a class-predictive MNIST color, a class-predictive texture pattern, or by stacking two datasets like MNIST and fashion-MNIST together. It would also be good to make it clear in the definitions that it is assumed that the spurious feature is not perfectly correlated with the label, and that this is explicitly different from those existing settings. Given that the "shortcut" definition used here substantially deviates from the existing literature, it may be useful to consider using a different term (since "shortcut learning" already has a precise, yet different, definition: a decision rule that generalizes IID but not OOD). (On a sidenote, I also found it confusing in Figure 1 right that the setting visualizes "testing data w/o shortcut" while a spurious feature s influences the data generation of x.)

---

> > > > > ### Author Response · Authors · 2022-12-12
> > > > > **Thank you for your last reply!**
> > > > >
> > > > > As mentioned above, We are in agreement:
> > > > > > If the only available data is that s and y are fully correlated, then we can see how the difficulty level is the same.
> > > > >
> > > > > For those cases, we assume that
> > > > > > assume that $\Psi(X \rightarrow s)$ is given or computed from another dataset that is not fully correlated with $y$.
> > > > >
> > > > > If that is the case, the shortcut can be detected.  We will clarify that in the paper.
> > > > >
> > > > > Please note: We assume that in the test data, the causal link between $s$ and $y$ is broken; hence the test data is not IID from the training data (OOD wrt to that data). The whole point of the paper is that this is not sufficient as a point of concern for the generalizability of the decision rule, and we need a notion of difficulty.
> > > > >
> > > > > *If you are satisfied with our reply, we would be thankful if you update your score accordingly.*
> > > > >
> > > > > Thank you

---

> > > > > > ### Comment · Reviewer_5RsT · 2022-12-12
> > > > > > **Score**
> > > > > >
> > > > > > Thanks for confirming. Since this assumption seems problematic in the sense that it excludes many of the most widely used shortcut datasets, I have decided not to raise my score. That said, I do think the paper's clarity benefits from making the definitions explicit (as the authors have done during the rebuttal).

---

> ### Author Response · Authors · 2022-11-16
> **Precise Definitions, Defining "Easy to learn", Circular Argumentation (Part: 2/3)**
>
> > **[…] The approach of relying on "easy to learn" comes with a number of problems: now, one can no longer ask whether a dataset has a shortcut since the "easy to learn" definition is inherently tied to a specific model […]**
> >
>
> The “easy to learn” definition is tied to a specific model, which is exactly what we are trying to say. Sorry if this wasn’t clear in the paper. “Shortcut” is not just related to the dataset alone but is also closely tied to the model and the task. What is a shortcut for one model may not be so for another. Let us illustrate this with an example. Consider input  $X = \{x_1,x_2,…,x_n\}$ that can be viewed as both a vector or a set. Consider two models, $f_1$ which is a permutation-invariant function (like Deep Sets [1]), and $f_2$ is a coordinate-sensitive function (e.g., MLP). If the spurious feature is always in $x_1$, it can be a shortcut for $f_2$ but may not be for $f_1$ because $f_1$ is coordinate agnostic. In fact one can show [1] that $f_1 = \rho(\sum_i \phi(x_i))$  *for suitable transformations $\phi$* *and $\rho$*.
>
> Please look at our general comment (a common response to all reviewers), where we clarify and formalize these definitions. While the current literature mostly treats shortcuts as synonymous with spurious features, we mathematically show the difference b/w the two concepts.
>
> > **[…]** **the authors fall short of providing a convincing definition of "easy to learn” […]**
> >
>
> Sorry if our language appears imprecise, but hopefully, our general comment and modifications to Sec 3 (see highlighted text) clarify this well. In shortcut learning, “easy to learn” cannot be defined only as a function of the dataset but also depends on the model and the task. We have formalized the notion of spurious features, task difficulty, and shortcuts in section 3. These definitions show that shortcuts should **not** be viewed **only** as a dataset-specific attribute **but are also** model-dependent (as is common in current literature). Therefore, the task difficulty metric $\Psi$ is a function of the task, the data distribution, and the model, and PD is one of many metrics that satisfies all these properties. In very simple terms, we want to emphasize that not all spurious features are learned during training (for a given model). We use this premise to develop experiments to understand the nature of shortcut learning and detect dynamics associated with easy-to-learn spurious features.
>
> > **[…] It doesn't help that the authors aren't always using their own definition either: "First, we consider two datasets, one where the shortcut is easier than the core feature and another where the shortcut is harder. […]**
> >
>
> Thank you for pointing out the typo. We agree this has caused a lot of confusion. We have changed the text to: “First, we consider two datasets, one where the spurious feature is easier than the core feature and another where the spurious feature is harder.”
>
> > **[…] At points, the authors revert to visual intuition for defining "easy to learn" […] there are major problems when defining shortcuts […] without applying the necessary rigor in doing so consistently and with a convincing definition.**
> >
>
> We appreciate your focus on this point. To order the tasks in Sec 4.1 and 4.2, we use the mean PD (see Fig-4 caption and Sec 4.1 highlighted text). We have modified the text to make this clear. Also, we hope the modifications to Sec 3 and our general comment helps you understand the rigor and consistency of the definitions used.
>
> > **Circular argumentation: shortcuts are *defined* as easy to learn. […] then it is no surprise that the easy to learn features are learned easily.**
> >
>
> We respectfully disagree with this.
>
> 1. Shortcuts are not just easy features, but they must also be spurious features (Definition-3 of our general comment makes this clear).
> 2. Knowing whether a feature is spurious requires more information, which can come from domain knowledge (e.g., a radiologist knows chest drain is a spurious feature for Pneumothorax detection). This is consistent with many real-world applications of ML. Our method does not address the case when nothing is known about the spurious features and if there is no way to explain it to a human.
> 3. The metrics we describe indicate when and where one should be concerned about problematic shortcuts. Further analysis requires either a dataset from a new environment or potential extra dimensions that might be “nuisances” [7] or spurious features.

---

> ### Author Response · Authors · 2022-11-16
> **Reproducibility, Novelty, Post hoc Explanations, Proposition (V-information) (Part: 3/3)**
>
> > **Reproducibility: Poor. The main proof is in the appendix.**
> >
>
> The proof is lengthy, and it is unfortunately not feasible for us to include it in the main paper, given the page limit. The large set of experiments for empirical evaluation further limits the space for theory. We have, of course, made changes to the informal proposition in Sec 3 (see highlighted text) to provide more intuition to the readers about the relevance and theoretical contributions of this proposition.
>
> > **[…] No code has been submitted, which is less than ideal from a reproducibility perspective. […]**
> >
>
> Please take a look at our general comment. We provide a link to the anonymous Github repository in the paper (see Appendix-A.6), and other details (like python packages and version numbers) are also included to reproduce our results easily.
>
> > **Novelty: Limited. The paper starts with very simplistic settings (binary classification) […] However, in all cases, the shortcuts are known beforehand or even introduced by the authors themselves, […]**
> >
>
> As we pointed out earlier, this is the *main misunderstanding* of our paper.  The latent variable is **not** used during the training. We only use it for evaluation. Please let us know where we can improve our writing so that this is clarified further. We respectfully disagree with the comment regarding the realism of the dataset. MIMIC-CXR and NIH datasets are some very realistic datasets from the medical domain.
>
> > **[…] if GRAD-CAM can show the shortcut, why do we need an additional method on top?**
> >
>
> We are **not** using gradCAM to detect the shortcut. It is **not** recommended to rely solely on viz techniques (like GradCAM, saliency maps, etc.), as confirmed by several previous works [2,3,4]. Our approach is not a post hoc explanation, but rather we suggest adding a new quantity (e.g., PD) to be inspected during training (beyond training/validation loss).  If the reference PD plots are given, and humans have an intuition about the level of difficulty of a task (similar to Fig 4), no visualization is needed. When there is no reference plot, then our approach can still be used to monitor suspicious learning trends. On observing such a behavior, we suggest that the user stops training and analyzes the samples located in the spurious peaks (either by manual inspection or visualization techniques like GradCAM or histogram/AUC analysis, as shown in Fig-6 NIH experiments).
>
> > **Furthermore, the proposition (connection to V-information) seems close to obvious if I'm not mistaken: if one introduces more information, there's more information for the model to pick up on.**
> >
>
> We're sorry if the proposition was not fully clear. We believe this is again due to the misunderstanding that the spurious feature $s$ is known during training. $s$ is a latent variable, so it is not given during training, which is why the proposition is not obvious or apparent. Additionally, our focus is on $\mathcal{V}$-information or usable information, not Shannon Information (or mutual information). $\mathcal{V}$-information depends on the model class and the type of features being used, unlike Shannon Information which is independent of these and assumes unbounded computation. So unless the model can express the relationship between the label and the spurious feature, the model cannot use it. So we respectfully disagree with your claim that introducing more information means the model has more information to pick up on. This is false because we can add information that is not *******usable******* or learnable by networks of a fixed depth [5,6]. Adding such information (non-usable) will not make any difference to the model. Moreover, PD and PVI are two very recent metrics proposed for measuring instance difficulty, and the theoretical (or empirical) connection between $\mathcal{V}$-information and average PD has not been shown or investigated previously. Furthermore, the proposition helps justify the empirical success of our approach to detect shortcut learning.
>
> *Please let us know if we have clarified all your questions and if this convinces you about our hypothesis and experiments. We are happy to discuss this further with you if needed.*
>
> **References:**
>
> * [1] Manzil Zaheer, et al. "Deep sets." NIPS (2017).
>
> * [2] Adebayo, Julius, et al. "Post hoc explanations may be ineffective for detecting unknown spurious correlation." *ICLR* 2021.
>
> * [3] Adebayo, Julius, et al. "Sanity checks for saliency maps." NIPS (2018).
>
> * [4] The (Un)reliability of saliency methods. (NIPS 2017) Workshop submission
>
> * [5] Ethayarajh, Kawin, et al. "Information-theoretic measures of dataset difficulty." (2021).
>
> * [6] Xu, Yilun, et al. "A theory of usable information under computational constraints." (2020).
>
> * [7] Puli, Aahlad Manas, et al. "Out-of-distribution Generalization in the Presence of Nuisance-Induced Spurious Correlations." *ICLR (2021)*

---

### Official Review · Reviewer_eVSL · 2022-10-25

**Confidence:** 4
**Correctness:** 3
**Technical Novelty And Significance:** 2
**Empirical Novelty And Significance:** 3
**Recommendation:** 6

**Clarity, Quality, Novelty And Reproducibility:**

The writing is clear and concise, which is appreciated.  While the authors do not provide any precise definition of difficulty, they do establish convincingly that shortcuts are more aligned with 'easily detectable spurious correlations' rather than difficult ones.

The authors demonstrate their claim quite effectively with a well-constructed hybrid (or "dominios" in the parlance of Kirichenko et al.) datasets that couple an unaltered dataset (Fashion MNSIT) with low-PD shortcuts (MNIST) and high-PD shortcuts (CIFAR10).  I find it particularly compelling not only based on the drastic Table 1 figures, but also that CIFAR10 is not considered a particularly difficult task for a standard ResNet18.

The primary novel contribution here is the realization that PD can be used to interrogate models to detect the presence of undesirable shortcut learning, rather than merely to stratify a dataset by degree of difficulty.   The authors use publicly available data, and standard models for their experiments, rendering the work more reproducible (modulo any code to be released upon decision).


**Strength And Weaknesses:**

### Strengths
- The experiments are well-designed.  Reading sections 4.1 through 4.4, I could not help but think of follow-on problems to examine for each, probing the fidelity of prediction depth as a function of number of parameter updates, as well as of the relative abundance or scarcity of the data.
- The figures (2-7 certainly) are very clearly designed, and informative
- The direct manner in which the authors lay out their premies in section (1) makes the claims of the paper simple to evaluate in light of the evidence they present.


### Weaknesses
- The authors do not provide any precise definition for easy tasks versus hard tasks, leaving the reader to intuit why certain tasks are ordinally related in difficulty (e.g the claim in section 4 that MNIST is simpler than Fashion MNIST, which in turn is simpler than KMNIST).  Spending a bit more time to agree on a way to measure degrees of difficulty would help not only this paper, but would help to broaden the applicability of their method to other domains.  Perhaps some measure of generalization?

(section 1) Panel (B) of Figure 1 has no caption text; is this intentional or is it an oversight?
(section 4.1) A few suggestions for strengthening this experiment:
1. The distribution (or first two moments) over several re-trainings should be reported in table 1.  It's always better to report distributions rather than point estimates, especially when the point estimate is not precisely defined (i.e was only one experiment performed?  Or does table 1 contain the max / min disparity?)
2. The results in Table 1 are conditioned on the difficulty of the binary classification tasks, so the authors should either report the difficulty of each of the pairwise core-feature tasks, or try multiple core-feature versus spurious feature combinations to better characterize the shortcut effect.

**Summary Of The Paper:**

This paper is concerned with clarifying the distinction between shortcut features and spurious correlations.  The authors note that while spurious correlations are rightly acknowledged as a source of shortcuts, and thus for shortcut learning to be identified with distribution shifts, there is a distinct source of shortcuts that is based on the difficulty of shortcut features.  They investigate the learning of shortcut features by interrogating training dynamics and example difficulty, positing that easy features learned by the initial layers of a DNN  early in training are potential shortcuts.  They provide empirical evidence for this hypothesis with synthetic and real medical imaging data, and they show that  prediction depth and $\nu$-usable information are correlated.

In summary, the authors establish that:
1. Shortcut features are spurious correlations, but not all spurious correlations are shortcuts
2. The presence of shortcut features may sometimes be detected by looking at the distribution of prediction depths of a validation set (or test set)
3. The readily computable prediction depth is positively correlated with $\nu$-usable information, a refinement of mutual information that takes computability constraints into account.


**Summary Of The Review:**

The authors succeed in convincing the reader that spurious correlations are not always shortcuts, though I would argue that they omit connecting their work to the growing literature on spurious correlations.  Perhaps this is intentional, to limit the scope of their claims, but I feel it could be a missed opportunity.  For example, recent work by [Veitch, D'Amour, Yadlowsky and Eisenstein](https://arxiv.org/abs/2106.00545) considers spurious correlation detection using tools from causality, where they distinguish different model features as being independent to changes in any latent factor that may introduce causal (or anti-causal) links to the labels. The authors herein may want to follow up on that line of work to characterize the 'easy' features they detect directly via prediction depth.

While the claims of novely may be modest, it is clearly presented and may open up further work to understand shortcut learning through spurious correlations.

---

> ### Author Response · Authors · 2022-11-16
> **Precise Definitions, Multiple Core-Spurious Combinations (with several re-trainings),  Literature on Spurious Correlations**
>
> Thank you for the detailed feedback and wonderful suggestions. We are glad you find our work *well-designed*, *clear*, and *informative*. Please see the below responses to your queries:
>
> > **The authors do not provide any precise definition for easy tasks versus hard tasks […]**
> >
>
> Sorry for the confusion. We should have made this clear in the paper. We ordinally relate the various tasks using the *mean PD* obtained on these datasets (see Fig-4 and highlighted caption). We have modified Sec 4.1 (please see highlighted text) to clarify this. Given the literature on task difficulty and instance difficulty metrics, one can easily obtain an approximate estimate of the difficulty of a dataset. Fig-4 shows the mean PD (dotted vertical line) for the various datasets (MNIST=2.2, FMNIST=3.9, CIFAR10=6.8), which results in the following order: MNIST $<$ FMNIST $<$ CIFAR10.
>
> We also agree we have been a bit imprecise in our usage of terms like task difficulty, spurious features, etc. As mentioned in our general comment above, we formalize and make the definitions more concrete by adding mathematical definitions for the various concepts involved in the paper (see highlighted text in Sec 3).
>
> We agree that some measure of generalization will help broaden the method's applicability. We will consider this as part of our future work. Thank you.
>
> > **(section 1) Panel (B) of Figure 1 has no caption text; is this intentional or is it an oversight?**
> >
>
> Sorry for the confusion. The test images do not have spurious features. We have added a caption to make this clear.
>
> > **The distribution (or first two moments) over several re-trainings should be reported in table 1 […]**
> >
>
> Thank you. We agree! We have updated Table 1 to report the mean and standard deviation over four re-trainings. The results are no longer point estimates. We find the results and our observations to be consistent across re-trainings.
>
> > **The results in Table 1 are conditioned on the difficulty of the binary classification tasks […] try multiple core-feature versus spurious feature combinations to better characterize the shortcut effect.**
> >
>
> Thank you again for the great suggestion! As mentioned above, we have updated Table 1 to include six domino-dataset (core-spurious) combinations to strengthen our claim that “Not all spurious correlations are shortcuts.” We consistently see that when the spurious feature is harder than the core feature, it is not used as a shortcut by the model. Only those spurious features that are easier than core features end up becoming shortcuts.
>
> > **[…] I would argue that they omit connecting their work to the growing literature on spurious correlations. […] For example, recent work by [Veitch, D'Amour, Yadlowsky and Eisenstein](https://arxiv.org/abs/2106.00545) considers spurious correlation detection using tools from causality […]**
> >
>
> That’s a very nice suggestion! We have added this reference to the related work section and will try to follow up on this line of thought (e.g., characterizing shortcuts and spurious features using tools from causal inference) in the future.
>
> *Please let us know if we have addressed all your concerns and if the additional experiments have further convinced you about the empirical evaluation of our method and hypothesis. We are happy to provide further clarification.*

---

> > ### Comment · Reviewer_eVSL · 2022-11-22
> > **Responses to authors**
> >
> > First of all, thanks for taking the time to address my questions.  I especially appreciate the effort to specify definitions of easy versus hard tasks, and for emphasizing the mean PD ranking of datasets in section 4.1.
> >
> > I'm a bit disappointed with the additional contribution of appendix A.8.  The authors carry out an experiment therein to look at the mean conditional $\mathcal{V}$-entropy of different datasets for examples by interval-stratified prediction depth (Figure 15).  While Figure 15 aligns with their previous claim that $\mathcal{V}$-information and prediction depth are correlated, it seems that they could have gotten some more valuable insight with a bit more effort.
> >
> > For example, the differences in the height of each of the steps is worth remarking upon.  Back in section 4.1, the authors order these datasets by increasing mean PD.  Figure 15 shows that this ordering hides a lot of structure in the data, with SVNH having a mixture of very simple and very difficult problems, and KMNIST even more so.  Also, the presentation of Figure 15 makes me wonder what is the relationship between the observed conditional entropy of the models (the softmax entropy) in each of these grouped PD subsets.  Would taking the point estimates of each of the datapoints be sufficient to stratify them into their PD categories (which would be much cheaper to compute)?  I doubt that such a simple relationship exists, but it's worth disproving.
> >
> > All taken, I appreciate the authors efforts to clarify their core definitions and to expand the scope of their experiments to other vision datasets.  I remain convinced that it's a small but worthwhile contribution to the literature, and deserves to be published.

---

### Official Review · Reviewer_RFyP · 2022-10-25

**Confidence:** 3
**Correctness:** 3
**Technical Novelty And Significance:** 3
**Empirical Novelty And Significance:** 3
**Recommendation:** 6

**Clarity, Quality, Novelty And Reproducibility:**

Please see above for my evaluation of the clarity, quality and novelty of the paper. While the clarity and novelty are undoubtedly very good, the quality (empirical validation) on the other hand can be improved.

Regarding reproducibility, the submission didn’t include anonymized source code. However, given the overall simplicity of the paper, I believe it won’t be too hard to reproduce its results even without the source code.

Below are some additional questions for the authors.
* The authors conjecture that “easy features learned by the initial layers (…) are potential shortcuts”. If so, will simply freezing the early layers at some known good pretrained weights solve the problem entirely? Or will the shortcuts manifest again in a different way (e.g. dynamics) at later layers?
* Prior work suggests that post hoc explanation methods can be ineffective when the spurious features are unknown to the user at test time (e.g. non-visible artifacts, undertraining, etc.) [5]. Is the proposed workflow also vulnerable to this issue, or does it provide additional guarantees to mitigate or eliminate this issue?

[5] Post hoc Explanations may be Ineffective for Detecting Unknown Spurious Correlation, ICLR, 2022.

**Strength And Weaknesses:**

Strengths
+ (Clarity) The paper is well organized and clearly written. Fig 1 (prime number vs white patch) is a great example explaining the paper’s main finding.
+ (Novelty) The paper’s main finding, effective tool usage (PD) and theoretical analysis (connection between PD and PVI) are all novel contributions as far as I know.
+ (Significance) Given the importance of the topic, the novelty of the contributions, and the overall effectiveness of the proposed workflow (although can be better validated, see below), I believe this paper can be a strong, potentially impactful baseline for future researchers & practitioners to use and/or improve upon.

Weaknesses
- (Quality) While the paper’s technical details are all correct as far as I can tell, the empirical validation still can be improved in the following aspects.
1. Although medical imaging indeed is a safety-critical domain and deserves more studies regarding spurious correlations, having only medical (chest X-ray) datasets with real (non-synthetic) shortcuts in the experiments is unfortunately non-ideal. Please consider including other commonly used datasets e.g. CelebA (gender vs hair color), Waterbirds (bird type vs background) [1] and/or NICO (object vs context) [2, 3] to diversify and strengthen the empirical evaluation.
2. Please consider strengthening the claim that only “easier” spurious features can be shortcuts with larger-scale experiments, such as a) randomly combining (as Sec 4.1) more datasets with different PD difficulties or b) creating & using synthetic datasets with controllable shortcut difficulties like e.g. [4], in order to quantify the correlation between feature difficulties and utilization (via e.g. GradCAM analysis).
3. Similarly, it would be a plus to showcase the correlation between PD and PVI in Sec 4.4 with larger-scale experiments to empirically support Proposition 1 more solidly.

[1] Distributionally robust neural networks for group shifts: On the importance of regularization for worst-case generalization, ICLR, 2020.\
[2] Towards non-iid image classification: A dataset and baselines, Pattern Recognition, 2021.\
[3] NICO++: Towards Better Benchmarking for Domain Generalization, https://arxiv.org/abs/2204.08040 \
[4] A Fine-Grained Analysis on Distribution Shift, ICLR, 2022.

**Summary Of The Paper:**

The paper studies the relationship between spurious features, shortcuts and learning dynamics in a novel way. By effectively using the prediction depth (PD, Baldock et al.) as its main tool, the paper concludes that only spurious features that are “easier” than core features (instead of all spurious features) can be shortcuts, and the learning of such shortcuts can be identified early during training via PD and GradCAM visualization. The proposed workflow is validated on both synthetic and real (chest X-ray) datasets and shown effective in spotting the shortcuts. Additionally, the paper theoretically connects PD and usable information (PVI, Ethayarajh et al.) to explain its empirical success.

**Summary Of The Review:**

The paper in its current form has multiple great strengths (clarity, novelty, significance) but also some unignorable weaknesses (empirical validation). In my opinion, since the paper’s strengths outweigh its weaknesses, I’m learning towards recommending its acceptance to the conference.

---

> ### Author Response · Authors · 2022-11-16
> **Reproducibility, Post hoc Explanations, Transfer Learning (Part: 1/2)**
>
> Thank you for your wonderful suggestions on improving the empirical evaluation of our method! We are currently running and compiling results for the experiments you requested for. Please give us a few days to get back to you. Below are responses to your other comments:
>
> > **Regarding reproducibility, the submission didn’t include anonymized source code […]**
> >
>
> We are sorry for the delay. Please take a look at our general comment. We provide a link to the anonymous Github repository in the paper (see Appendix-A.6), and other details (like python packages and version numbers) are also included to reproduce our results easily.
>
> > **[…]** **post hoc explanation methods can be ineffective when the spurious features are unknown […] Is the proposed workflow also vulnerable to this […]”**
> >
>
> That’s a very good point! Our approach is not a post hoc explanation, but rather we suggest adding a new quantity (e.g., PD) to be inspected during training (beyond training/validation loss). If the reference PD plots are given, and humans have an intuition about the level of difficulty of a task (similar to Fig-4), no visualization is needed. When there is no reference plot, then our approach can still be used to monitor suspicious learning trends. On observing such a behavior, we suggest that the user stops training and analyzes the samples located in the spurious peaks (either by manual inspection or visualization techniques like GradCAM or histogram/AUC analysis, as shown in Fig-6 NIH experiments).
>
> > **[…] will simply freezing the early layers at some known good pretrained weights solve the problem entirely? […]**
> >
>
> That’s a great point! Our intuition strongly suggests that it won’t solve the problem fully. We believe that deep learning models, owing to their high capacity, will pick up the shortcut using the later layers. Transfer learning is similar to your suggestion and does help to some extent with shortcut learning, but it doesn’t solve the problem fully. We are happy to investigate this further, and if time permits will get back to you with preliminary evidence in this regard.
>
> Currently, we have prioritized running experiments for your other major comments (namely: (1) including vision datasets like NICO, (2) empirically strengthening the claim in Sec 4.1, (3) Correlation b/w PD and PVI in Sec 4.4)

---

> ### Author Response · Authors · 2022-11-19
> **Strengthening and Improving the Empirical Evaluation (Part: 2/2)**
>
> Thank you for your valuable suggestions, which helped us considerably improve our empirical validation. We have performed the experiments you asked for.
>
> > **[…] Please consider including other commonly used datasets e.g. […] NICO (object vs context) [2, 3] to diversify and strengthen the empirical evaluation.**
> >
>
> Please see Appendix-A.7 (Vision Experiments). We use the NICO++ dataset to create multiple spurious datasets (Cow vs. Bird, Dog vs. Lizard, Flower vs. Lizard) such that the context/background is spuriously correlated with the target. We test our hypothesis using ResNet-18 and VGG16.
>
> Our experiments (see Figures 11,13,14) consistently show that models that learn spurious correlation exhibit PD plots skewed toward the initial layers. This indicates that the model relies heavily on very simple (potentially spurious) features for the task. GradCAM maps also confirm that the model often focuses on the background context rather than the foreground object of interest.
>
> We further observe (see Fig-12) that balancing the training data (to remove the spurious correlation) results in a model with improved test accuracy (80.2%) as expected. This is also captured by the PD plot (Fig-12B), where we see that the distribution of the peaks, as well as the mean PD, shifts proportionately towards the later layers, indicating that the model now relies lesser on the spurious features.
>
> By monitoring PD plots during training and using suitable visualization techniques, we show how
> one can obtain useful insights about the spurious correlations that the model may be learning. This
> can also help the user make an educated guess about the generalization behavior of the model during deployment.
>
> *Our experiments (both medical and vision) show the general nature of our hypothesis and method.*
>
> > **Please consider strengthening the claim that only “easier” spurious features can be shortcuts with larger-scale experiments […]**
> >
>
> Thank you again for the suggestions! Please take a look at Section-4.1 and the results in Table 1. We now show results on *six* *datasets* with spurious-core combinations of varying difficulty. We construct three pairs of domino datasets such that each pair has both a hard and an easy spurious feature with respect to the common core feature. We also show the individual dataset difficulties (measured using mean PD) to quantitatively compare spurious and core datasets and test our hypothesis.
>
> All the results in Table 1 consistently support the claim that only those spurious features that are ******easier****** than core features are leveraged by the model as shortcuts (indicated by a significant drop in core-only accuracy).
>
> *We hope this further convinces you about the premises that led to our hypothesis.*
>
> > **Similarly, it would be a plus to showcase the correlation between PD and PVI in Sec 4.4 with larger-scale experiments to empirically support Proposition 1 more solidly.**
> >
>
> Please refer to Appendix-A.8, where we empirically validate the relationship between PD and $\mathcal{V}$-information. We use VGG16 trained on *************four datasets************* (KMNIST, FMNIST, SVHN, CIFAR10) to measure the correlation between PD and $\mathcal{V}$-information. The results consistently show that these two quantities are positively correlated. Instance difficulty increases with PD, and the usable
> information decreases with an increase in V-entropy. It is, therefore, clear from Fig-15 that samples
> with a higher difficulty (PD value) have lower usable information. This also provides empirical support to Proposition-1 in Appendix-A.1.
>
> *Please let us know if we have addressed all your doubts and if the above experiments further convince you about the empirical validation of our method. We are happy to provide more clarification if needed.*

---

> ### Comment · Reviewer_RFyP · 2022-11-23
> **Re: Feedback**
>
> I appreciate the additional results and clarifications provided by the authors. While I can see the current revision being accepted to the conference, I also believe there’s still quite some room for more improvements.
> * Although the NICO++ results indeed diversify the empirical validation, I’m not sure if they substantially strengthen the paper. Firstly, the spurious features of the chosen splits (i.e. backgrounds) are mostly too easy to drive the PD peaks beyond the first layer (unlike e.g. the chest drains) to really test the proposed workflow’s effectiveness in identifying all kinds of shortcuts. According to Kirichenko et al., on the Waterbirds dataset, “NNs do not appear to be particularly biased towards either the background or the foreground, and treat them equally” based on their regular & inverted Waterbirds results.* The authors could consider using those Waterbirds settings (regular & inverted) or more challenging splits from NICO++. Secondly, the NICO++ results are not sufficiently analyzed either. E.g. in Fig 14, flowers (foreground, non-spurious) seem to be correctly picked up during training. In this case, why is the test accuracy still so poor and even worse than Fig 11 & 13?
> * The updated Table 1 is a nice improvement, but even more detailed analysis would be helpful. For example, as the MN-FMN and MN-KMN runs mostly have poor core-only accuracies (due to easy shortcuts), why do the KMNpatch-MN runs have (some) considerably better results? Is this due to the smaller difference between KMNpatch’s & MN’s PD difficulties (2.2-1.1 < 3.9-2.2 < 5-2.2)? More generally, how does the difference between PD difficulties affect the shortcut learning?
> * Fig 15 should be instead drawn as 2D scatter plots (where each dot marks one image’s PD and conditional V-entropy) to better visualize the details (as also asked by reviewer eVSL).
>
> *Kirichenko et al. also stated that “while it is often suggested that NNs are biased to learn the background, we see that in fact the network relies on the spurious foreground feature (bird) when trained to predict the background.” Does this contradict the paper’s claim? Assuming both their regular & inverted settings use the same foregrounds and backgrounds (thus unchanged difficulties), why do the spurious features seem to switch between the two tasks?

---

> > ### Author Response · Authors · 2022-12-11
> > **Further Analysis and Explanations for Table-1 and NICO++ results (Part-1/2)**
> >
> > We apologize for the delay. Please find our response below.
> >
> > > **[…] Firstly, the spurious features of the chosen splits (i.e. backgrounds) are mostly too easy [...] to really test the proposed workflow’s effectiveness in identifying all kinds of shortcuts. […]**
> > >
> > - We agree that the spurious features we chose are easy and can produce a peak in the first layer. While PD may not work as effectively for more challenging shortcuts, this is more of a limitation of PD than the proposed workflow. Our workflow is generic and not limited to PD. It is defined for any task difficulty metric $\Psi$. We simply show how monitoring the histogram of images with respect to $\Psi$ during training can lead to valuable insights in terms of shortcut learning and its detection. The better the techniques used (both $\Psi$ and visualization techniques), the better our proposed workflow will be able to detect shortcut learning.
> >
> > - NICO++ splits, in particular, have a few issues:
> >     - The dataset has a significant label noise (e.g., Images with the label “lizard on grass” had images of lizards on rocks, and vice versa). Disentangling the effects of label noise and shortcuts was challenging. This also resulted in PD plots being noisy.
> >     - NICO++ comprises high-resolution real images with a lot of diversity. The more realistic and diverse the dataset is, the more the sensitivity of PD plots with respect to hyperparameters. This is a limitation of PD.
> >
> > - Below is a table to compare a few task difficulty metrics across factors relevant to our work (scale of 1-5 where five is good/ideal, and one is poor)
> > | Task Difficulty Metric $\Psi$| Reliability on intermediate checkpoints | Sensitivity to hyperparameters | Consistency across random seeds and architectures | Meaningful and robust notion of task difficulty |
> > | --- | --- | --- | --- | --- |
> > | PD | 4 | 2 | 3 | 3 |
> > | PVI | 3 | 4 | 4 | 4 |
> > | VOG [2] | 2 | 2 | 3 | 2 |
> >
> > We use PD as it provides meaningful insights even on the intermediate checkpoints, which is useful for detecting shortcut learning early during training.
> > > **[…] why do the KMNpatch-MN runs have (some) considerably better results? [...] More generally, how does the difference between PD difficulties affect the shortcut learning?**
> > >
> >
> > Thank you for the good questions. The choice of features that the model chooses to learn depends on the PD distributions of the core and spurious features. We provide three different perspectives on why KMNpatch-MN runs have better results.
> >
> > **PD Distribution Perspective:** The KMNpatch-MN domino dataset has a smaller difference in the core-spurious mean PDs (as you rightly observe). The closer the PD distributions of the core and spurious features are, the more the model treats them equivalently. Therefore, in the case of the KMNpatch-MN, we empirically observe that different initializations (random seeds) lead to different choices the model makes in terms of core or spurious features. This is why the standard deviation of KMNpatch-MN is high (20.03) compared to the other experiments (~1).
> >
> > **Theoretical Perspective (Proposition-1):** This is not surprising and, in fact, corroborates quite well with the Proposition-1 in Appendix-A.1. The *Prediction Depth Separation Assumption* suggests that without a sufficient gap in the mean PDs of the core and spurious features, one cannot concretely assert anything about their ordinal relationship in terms of their $\mathcal{V}$-usable information. In other words, spurious features will have higher usable information (for a given model) than the core features only if the spurious features have sufficiently lower mean PD as compared to the core features. On the other hand, as the core and spurious features become comparable in terms of their difficulty, the model begins to treat them equivalently.
> >
> > **Loss Landscape Perspective:** (this is just a conjecture) The loss landscape is a function of the model and the dataset. The solutions in the landscape that are reachable by the model depend on the optimizer and the training hyperparameters. Given a model and a set of training hyperparameters, we conjecture that the diversity of the solutions in the landscape increase as the distance (difference in mean PD) between the core and spurious features decreases. This diversity manifests as the model's choice of using core vs. spurious features and could potentially result in a higher standard deviation of core-only accuracy across initializations.

---

> > ### Author Response · Authors · 2022-12-11
> > **Further Analysis and Explanations for Table-1 and NICO++ results (Part-2/2)**
> >
> > > **Kirichenko et al. also stated that “while it is often suggested that NNs are biased to learn the background, we see that in fact the network relies on the spurious foreground feature (bird) when trained to predict the background.” Does this contradict the paper’s claim? […]**
> > >
> >
> > That’s a very good observation! Those findings do not contradict our paper’s claim. Please see the part-1 response to your question (regarding why KMNpatch-MN runs have better results). Without a good *prediction depth separation* between the core and spurious features, we know (Proposition 1, empirical observation) that the model treats the core and spurious features equivalently. This is perhaps why both the regular and the inverse waterbird problem have identical results.
> >
> > > **Secondly, the NICO++ results are not sufficiently analyzed either. E.g. in Fig 14, flowers (foreground, non-spurious) seem to be correctly picked up during training. In this case, why is the test accuracy still so poor and even worse than Fig 11 & 13?**
> > >
> >
> > That's a good point; thank you! We have analyzed the results in more detail now:
> >
> > 1. The test accuracy is poor as the model is looking for the white color and not flowers. The training data is largely comprised of white-colored flowers (~50%), and hence the model uses white color to detect flowers. The test data has nearly equal proportions of different colored flowers, so the test accuracy is poor.
> > 2. To verify the above statement, we perform counterfactual analysis by measuring the individual and average treatment effects (ITE and ATE in causal inference). The model has an accuracy of \~70% (\~50%) for white (non-white) flowers. This 20% decrease led to a drop in the overall test accuracy. We also see a 10% drop in accuracy when we change the color of white flowers to a different random color (using image processing).
> >
> > (Note:  Visualization techniques like saliency maps or GradCAM can only answer questions like “where is the model looking?” but not questions like “what is the model looking for?” or “why is the model looking at this region?”. To answer the “what” and “why” questions requires techniques like counterfactual explanations [1].)
> >
> > 3. To conclude, our method detects suspicious activity during training. Knowing the cause for this behavior is beyond the scope of the current paper and requires more investigation, as shown above.
> >
> > > **Fig 15 should be instead drawn as 2D scatter plots (where each dot marks one image’s PD and conditional V-entropy) to better visualize the details (as also asked by reviewer eVSL).**
> > >
> >
> > Thank you! PD is a discrete quantity, whereas conditional V-entropy is continuous. Therefore a simple 2D scatter plot may not be appropriate. Instead, we will update the paper with the distribution of the conditional V-entropy values for each of the stratified PD bins.
> >
> > ********************References********************
> >
> > - [1] Cohen, Joseph Paul, et al. "Gifsplanation via latent shift: a simple autoencoder approach to counterfactual generation for chest x-rays." *Medical Imaging with Deep Learning*. PMLR, 2021.
> > - [2] Agarwal, Chirag, Daniel D'souza, and Sara Hooker. "Estimating example difficulty using variance of gradients." CVPR 2022.

---

> > ### Author Response · Authors · 2022-12-12
> > **Final Comment**
> >
> > *If you are satisfied with our reply, we would be thankful if you update your score accordingly.*

---

### Official Review · Reviewer_3xDy · 2022-10-25

**Confidence:** 2
**Correctness:** 3
**Technical Novelty And Significance:** 3
**Empirical Novelty And Significance:** 4
**Recommendation:** 6

**Clarity, Quality, Novelty And Reproducibility:**

Clarity:
The presentation (section 3) and the proof (A.1) of the relationship between prediction depth and V-information should be improved.

Novelty and Quality:
The method of using the PD in order to detect that a trained network is relying on spurious correlations is novel and relevant as far as I know. The experiments on medical images are convincing.

**Strength And Weaknesses:**

Strength:
 - The idea of using the PD in order to identify potential spurious correlations is new and interesting. The experiments on toy and medical images provide interesting new insights.

Weaknesses:
 - It feels like the statement "not all spurious correlations are shortcuts" is somewhat trivial
 - I was not able to fully grasp the theoretical contribution in section 3. Arguably, this could be because I am missing some background on the subject, but I think that this also means that the presentation of your proposition 1 could be improved, even if still informal.

Other suggestions:
Figure 6: I suggest also looking at epoch 0 or in other words at initialization before any training.
Section 4: "shorcut is harder" why is it a shortcut then ?

**Summary Of The Paper:**

Through an empirical study on vision benchmarks and medical images, this paper explores the concepts of spurious and shortcut features and their links to prediction depth and V-information. They show empirically that shortcuts can be detected early in training. They also show a link between prediction depth and V-information.

**Summary Of The Review:**

I liked the method of using the PD in order to examine training and identify potential spurious correlations. I think that the paper could be improved by clarifying the theory part.

---

> ### Author Response · Authors · 2022-11-16
> **Precise Mathematical Definitions, Proposition-1, Epoch-0 PD Plots**
>
> > **It feels like the statement "not all spurious correlations are shortcuts" is somewhat trivial.**
> >
>
> Thank you for your valuable feedback. This helps us improve the definition of the shortcut in our paper. Currently, in the literature, spurious features are defined in terms of a distribution shift (between train and test) and treated as synonymous with shortcuts [1-3]. We aim to provide a more rigorous definition of shortcuts beyond the distribution-shift perspective. More specifically, our definition of shortcut depends **not only** on data **but also** on the model and whether it is learnable by the model. To the best of our knowledge, such a model-dependent definition of shortcut has not been formalized in the literature. As mentioned in our general comment above, we have inserted mathematical definitions for various concepts (spurious features, task difficulty, shortcut) in Sec 3 (see highlighted text) to improve the paper's clarity.
>
> > **[...] presentation of your proposition 1 could be improved, even if still informal [...].**
> >
>
> Thank you for your suggestion! The general idea is as follows: $\mathcal{V}$-information is an information-theoretic approach to define task difficulty [4] that is not only dependent on data but also the model. This is in line with definition-2 above for task difficulty $\Psi$. Similarly, PD can also be used to measure task difficulty [5]. While $\mathcal{V}$-information has its roots in information theory and is more rigorously defined than PD, we found PD to be more practical because it can monitor the model even during the early stages of training ($\mathcal{V}$-information is defined only at model convergence). So the proposition relates these two metrics giving us the best of both worlds. We have also improved the informal presentation of proposition 1 (see highlighted text in section 3).
>
> > **Other suggestions: Figure 6: I suggest also looking at epoch 0 or, in other words, at initialization before any training.**
> >
>
> That’s an interesting suggestion! We agree readers might be interested to know more about what happens at initialization and how PD plots look at epoch 0. We have added a brief discussion on PD plots at initialization in the appendix (see A.6). Due to space constraints, we couldn’t add these details to the main paper. In short, we expect most samples to be undefined at epoch 0 (shown by the red bar), and we see some additional noisy patterns in the plot. The model is not trained during initialization, and hence the weights and features are random vectors carrying little information.
>
> > **[…] Section 4: "shorcut is harder" why is it a shortcut then?**
> >
>
> Thank you for pointing out this typo! We have changed the text to: “First, we consider two datasets, one where the spurious feature is easier than the core feature and another where the spurious feature is harder.” As already mentioned, we’ve formalized the definitions of spurious features, shortcuts, task difficulty, etc., in section 3 (see highlighted text) to make these concepts clear and concrete.
>
> *Please let us know if this clarified all your questions and has convinced you more about our hypothesis, theory, and empirical evidence.*
>
> References:
>
> - [1] Robert Geirhos et al. (2020): Shortcut learning in deep neural networks. Nature Machine Intelligence
> - [2] Olivia Wiles et al. (2021): A fine-grained analysis on distribution shift.
> - [3] David Bellamy et al. (2022): A structural characterization of shortcut features for prediction. European Journal of Epidemiology
> - [4] Kawin Ethayarajh et al. (2022) "Understanding Dataset Difficulty with $\mathcal{V}$-Usable Information." *International Conference on Machine Learning*. PMLR.
> - [5] Robert Baldock et al. (2021): "Deep learning through the lens of example difficulty." *Advances in Neural Information Processing Systems.*

---

> ### Author Response · Authors · 2022-12-12
> **Final comment**
>
> *If you are satisfied with our reply, we would be thankful if you update your score accordingly.*

---

### Author Response · Authors · 2022-11-16
**General Comment about Reproducibility and Precise Definitions**

We thank all reviewers for their time and detailed feedback. We are glad that reviewers found our work *novel* and *interesting*, *well organized* and *clearly written*, *strong* and *potentially* *impactful* *baseline* for future researchers.

The revised manuscript is uploaded, and the changes in the text are highlighted. We have provided individual responses to each of the reviewers. Please take a look at our general comment addressing the common concerns below.

### **1. Reproducibility and Submission of Anonymized Source Code**

We provide a link to the anonymous Github repository in the paper (see Appendix-A.6), and other details (like python packages and version numbers) are also included to reproduce our results easily. As remarked by a few reviewers, it is very easy to reproduce these results, given that we use standard architectures (ResNet, DenseNet, VGG) and publicly available datasets (CIFAR10, MNIST, MIMIC, etc.). Our approach uses simple techniques like prediction depth and gradCAM, which are easy to implement and quite stable/insensitive to hyperparameter variations. We will further clean up the code for the ease of readers.

### **2. Precise Definitions (spurious feature, task difficulty, shortcut)**

We agree we could have been more precise in terms like task difficulty, shortcut features, etc.  To formalize this better and make the definitions more concrete, we have added the following mathematical definitions to the paper (please see highlighted text in Sec 3):

Let $P_{tr}$ and $P_{te}$ be the training and test distributions defined over the random variables $X$ (input), $y$ (label), and $s$ (***latent*** spurious feature).

**Definition-1 (Spurious Feature** $s$**):**   A latent feature $s$ is called spurious if it is correlated with label $y$ in the training data but not in the test data. Specifically, the joint probability distributions $P_{tr}$ and $P_{te}$ can be factorized as follows.

$P_{tr}(X,y,s) = P_{tr}(X|s,y) P_{tr}(s|y)P_{tr}(y)$

$P_{te}(X,y,s) = P_{tr}(X|s,y) P_{te}(s)P_{tr}(y)$
The variable $s$ appears to be causally related to $y$ but is not. This is shown in Fig-1 (main paper). We also need a notion of task difficulty. The difficulty of a task depends on the model and data distribution ($X, y$).

**Definition-2 (Task Difficulty $\Psi$):**   Let $\Psi_{\mathcal{M}}^{P}(X \rightarrow y)$ indicate the difficulty of predicting $X \rightarrow y$ for a model $\mathcal{M}$, where $X,y \sim P$. Consider a joint distribution $(X,y,t) \sim P$ for two tasks, $t$ , and $y$. Then, $\Psi_{\mathcal{M}}^{P}(X \rightarrow y) > \Psi_{\mathcal{M}}^{P}(X \rightarrow t)$ indicates that the task $X \rightarrow y$ is harder than $X \rightarrow t$ for a given model $\mathcal{M}$.

**Definition-3 (Shortcut):**   The spurious feature $s$ is a potential shortcut for model $\mathcal{M}$ iff $\Psi_{\mathcal{M}}^{P_{tr}}(X \rightarrow y) > \Psi_{\mathcal{M}}^{P_{tr}}(X \rightarrow s)$. In other words, given the input $X$, predicting spurious feature $s$ is easier for $\mathcal{M}$ than predicting the true label $y$.

### **3. Summary of the work**

- Shortcuts are not just easy features, but they must also be spurious features (Definition-3 of our general comment makes this clear).
- The metrics we describe indicate when one should be concerned about problematic shortcuts. Further analysis requires either a dataset from a new environment or potential extra dimensions that might be spurious features.
- Knowing whether a feature is spurious requires more information, which can come from domain knowledge (e.g., a radiologist knows that a chest drain is a spurious feature for Pneumothorax detection). This is consistent with many real-world applications of ML. Our method does not address the case when nothing is known about the spurious features and if there is no way to explain it to a human.

---

### Decision · Program_Chairs · 2023-01-20

**Decision:**

Reject

**Justification For Why Not Higher Score:**

While this paper presents some novel and interesting ideas for analyzing and understanding spurious correlations and shortcuts, the overall message is not compelling enough to merit publication at this time. Improvements to the storyline, definitions, breadth of experimentation, and other suggestions made by the reviewers could make this a more impactful paper.

**Justification For Why Not Lower Score:**

N/A.

**Metareview: Summary, Strengths And Weaknesses:**

This paper analyzes shortcut learning and spurious features, studying training dynamics and drawing connections to the concepts of prediction depth (PD) and V-usable information, concluding that only those spurious features which are “easier” than core features can be considered shortcuts, and moreover that the learning of these shortcuts happens early in training and can be observed via PD and the previously established visualization method known as Grad-CAM.

Generally, the reviewers found the paper to be mostly clearly written and the experiments to be well-designed. On its over all merits, however, the reviewers offered split opinions, with some reviewers appreciating the idea of using the PD in order to identify potential spurious correlations, and others raising concerns with some of the main definitions.

In the lengthy discussion, numerous questions arose, such as easy versus hard tasks and the shortcut definition, among many others. There was no single common concern, but a host of smaller concerns adding up to an overall lukewarm impression across the board. No reviewers felt that the current version of this paper had strong enough results to have a significant impact on the field. It seemed instead that the community would be better served with a version of this paper that incorporated further revisions, clarifications, and honing of the message.

**Summary Of Ac-Reviewer Meeting:**

Through the course of the AC-reviewer discussion, it became clear that there was not strong consensus regarding this paper's main strengths. One point of agreement was that most reviewers found value in utilizing PD to identify potential spurious correlations, but overall no reviewer felt that this paper was bound to have a strong impact on the community.

There was also not a single weakness that all reviewers highlighted. There was some discussion about unclear definitions, particularly the definition of shortcuts, which ultimately raised concerns about the technical clarity of the paper and the strength of the main narrative.